# In Vitro Cytological Responses against Laser Photobiomodulation for Periodontal Regeneration

**DOI:** 10.3390/ijms21239002

**Published:** 2020-11-26

**Authors:** Yujin Ohsugi, Hiromi Niimi, Tsuyoshi Shimohira, Masahiro Hatasa, Sayaka Katagiri, Akira Aoki, Takanori Iwata

**Affiliations:** Department of Periodontology, Graduate School of Medical and Dental Sciences, Tokyo Medical and Dental University (TMDU), Tokyo 113-8510, Japan; ohsugi.peri@tmd.ac.jp (Y.O.); kawakami.peri@tmd.ac.jp (H.N.); shimohira.peri@tmd.ac.jp (T.S.); hatasa.peri@tmd.ac.jp (M.H.); aoperi@tmd.ac.jp (A.A.)

**Keywords:** lasers, periodontal tissue, photobiomodulation, cell proliferation, gene expression

## Abstract

Periodontal disease is a chronic inflammatory disease caused by periodontal bacteria. Recently, periodontal phototherapy, treatment using various types of lasers, has attracted attention. Photobiomodulation, the biological effect of low-power laser irradiation, has been widely studied. Although many types of lasers are applied in periodontal phototherapy, molecular biological effects of laser irradiation on cells in periodontal tissues are unclear. Here, we have summarized the molecular biological effects of diode, Nd:YAG, Er:YAG, Er,Cr:YSGG, and CO_2_ lasers irradiation on cells in periodontal tissues. Photobiomodulation by laser irradiation enhanced cell proliferation and calcification in osteoblasts with altering gene expression. Positive effects were observed in fibroblasts on the proliferation, migration, and secretion of chemokines/cytokines. Laser irradiation suppressed gene expression related to inflammation in osteoblasts, fibroblasts, human periodontal ligament cells (hPDLCs), and endothelial cells. Furthermore, recent studies have revealed that laser irradiation affects cell differentiation in hPDLCs and stem cells. Additionally, some studies have also investigated the effects of laser irradiation on endothelial cells, cementoblasts, epithelial cells, osteoclasts, and osteocytes. The appropriate irradiation power was different for each laser apparatus and targeted cells. Thus, through this review, we tried to shed light on basic research that would ultimately lead to clinical application of periodontal phototherapy in the future.

## 1. Introduction

Periodontal tissue consists of “gingiva, periodontal ligament, cementum, and alveolar bone” [1]. Periodontal diseases cause a wide range of inflammatory conditions that affect the periodontal tissue, which could lead to loss of teeth and contribute to systemic inflammation [2]. Basic periodontal therapy eliminates etiological factors for periodontal disease and relieves inflammation in periodontal tissues [3]. Recently, periodontal therapy using a laser, “periodontal phototherapy,” has attracted much attention [4]. Many reports have been published on the methods recommended for periodontal therapy using lasers [5,6]. In addition, laser irradiation is also applied to treat pressure ulcers [7] and pain associated with temporomandibular dysfunction [8]. However, basic research related to clinical research is inadequate. We propose that it is necessary to understand the molecular biological effects of periodontal phototherapy for periodontal therapy. Photobiomodulation (PBM) is a treatment method based on research findings suggesting that irradiation with specific wavelengths of red or infrared light produces a wide range of physiological effects in cells, tissues, animals, and humans [9]. A previous study reported that the effect was a nonthermal process involving endogenous chromophores eliciting photophysical and photochemical phenomena at various biological scales, resulting in beneficial therapeutic outcomes [10].

The purpose of this review was to investigate cytological responses against laser irradiation for periodontal regeneration. We tried to provide insights on basic research that would subsequently lead to clinical research on periodontal phototherapy in the future.

## 2. Interaction with Tissues

When laser energy reaches a tissue surface, it can be reflected, scattered, absorbed, or transmitted to the surrounding tissues. The performance of a laser is determined by the degree of absorption. In particular, absorption in biological tissues is strongly influenced by the absorption coefficient in water, which is inherent to each wavelength [11,12]. Thus, lasers are clinically classified into two types depending on their wavelength: (1) a deeply penetrating type where the laser light penetrates and scatters into the tissue more deeply, such as the neodymium-doped yttrium-aluminum-garnet (Nd:YAG) (1064 nm) and diode lasers (810–980 nm available for clinical application and (2) a superficially absorbed type (shallowly penetrating type) where the laser light is absorbed in the superficial layer and does not penetrate or scatter deeply, such as the carbon dioxide (CO_2_) (10,600 nm), erbium-doped yttrium-aluminum-garnet (Er:YAG) (2940 nm), and erbium, chromium: yttrium –scandium-gallium-garnet (Er,Cr:YSGG) (2780 nm) lasers [4,13].

## 3. Effects of Laser Irradiation on Osteoblasts

The number of reports about the effects of laser irradiation on osteoblasts is increasing. For periodontal regeneration, osteoblasts play essential roles in bone formation and remodeling [14]. Therefore, the laser irradiation of osteoblasts is an important focus of research.

Most reports on laser-irradiated osteoblasts in vitro used diode lasers, including a blue diode laser (λ = 450 nm), a red diode laser (λ = 635–660 nm), and a Ga-Al-As laser (λ = 780–980 nm). Several studies have reported the effects of the Nd:YAG laser (λ = 1064 nm) on osteoblasts in vitro. A few reports on the effects of CO_2_ and Er:YAG lasers (λ = 2940 nm) have been published. 

### 3.1. Diode Lasers

Various kinds of osteoblasts or osteoblast-like cells were used in previous studies to assess the effects of diode lasers. Most studies used cell lines such as MC3T3-E1 cells, an osteoblastic cell line derived from mouse, in 12 reports [15,16,17,18,19,20,21,22,23,24,25,26]. The effects of laser irradiation on Saos-2 [27,28,29,30,31,32,33,34], MG-63 [17,35,36,37,38,39,40], and human osteoblastic cell lines [32,41,42,43,44,45,46,47] were investigated in 8, 7, and 8 studies, respectively. In addition, primary osteoblasts from rat calvaria or human bone were also used in 8 and 5 studies, respectively. 

Many studies have reported on the proliferation of cells irradiated by diode lasers. Diode laser irradiation significantly increased cell proliferation 1–3 days after irradiation [17,20,21,24,25,26,28,29,33,34,42,48,49,50,51]. Most of the effective energy density (fluence) ranges were from 1 to 10 J/cm^2^. In a previous study, irradiation at a total energy of 45.9–137.6 J/cm^2^ significantly increased the proliferation of human fetal osteoblasts (hFOB 1.19) [42]. Laser irradiation using various fluences (0.48–3.84, 5.0–8.3, and 45.9–137.6 J/cm^2^) also significantly enhanced cell proliferation at a later period of observation (e.g., on day 4–12) [29,42,48,52,53,54,55,56]. The proliferation of hypoxic-cultured osteoblasts was increased at 24 and 72 h after irradiation at 1.2 and 3.6 J/cm^2^, respectively [41]. In contrast, diode laser irradiation at similar fluences did not show a significant increase in osteoblast proliferation in some studies [17,27,35,38,57,58]. However, fluorescence-activated cell sorting (FACS) analysis of the cell cycle revealed that the percentage of cells in G_2_/M phase was significantly greater in rat calvarial osteoblastic cells by diode laser irradiation at 3.8 J/cm^2^ at 12 h after irradiation compared to nonirradiated control cells [49]. Cell viability and migration have also been evaluated in many studies. They were significantly increased by irradiation at 0.5–12 J/cm^2^ [30,32,36,43,44,45,47,56,59,60] Irradiation at fluences greater than 20 J/cm^2^ significantly decreased cell viability [30]. Previously, in most studies, it has been reported that diode laser irradiation at 1–12 J/cm^2^ tended to enhance the proliferation and viability of osteoblasts. However, effective irradiation protocols of diode lasers on the migration of osteoblasts have not been specifically determined.

Calcification of osteoblasts promoted by diode laser irradiation at 0.4–8.3 J/cm^2^ has been demonstrated in several studies [23,26,29,33,35,46,49,50,52,53,54,56,59,61,62,63]. Significantly enhanced mineralization in osteoblasts was observed at 7 days at the earliest [23], and in many cases, at around 20 days after irradiation [26,33,35,49,52,53,54,56,59,63]. Both single irradiation and multiple irradiations significantly promoted the calcification of osteoblasts.

Diode laser irradiation has been reported to affect gene and protein expression related to osteogenic differentiation, including alkaline phosphatase (ALP), osteocalcin, type Ⅰ collagen, Runt-related protein transcription factor 2 (Runx2), osterix, bone morphogenetic proteins (BMPs), transforming growth factor-β1 and β2 (TGF-β1 and β2), osteopontin, receptor activator of NF-κB ligand (RANKL), and osteoprotegerin (OPG).

At 1–14 days after irradiation, mRNA expression of ALP was significantly increased by irradiation at 0.4–6.7 J/cm^2^ [26,33,35,39,40,46,54,57]. ALP activity was also significantly enhanced at day 1–18 after irradiation at 1–10 J/cm^2^ in a number of studies [17,22,23,26,28,29,37,46,49,52,53,56,62,64]. Meanwhile, a previous study revealed that irradiation at 2 J/cm^2^ significantly decreased ALP activity in osteoblasts at 48 and 72 h [34]. Some studies have reported that diode laser irradiation did not increase ALP activity of primary osteoblasts after irradiation at 1.5 or 3 J/cm^2^ [57,58].

Osteocalcin, a marker of osteoblast terminal differentiation [65], has been investigated in relation to calcification of osteoblasts by diode laser irradiation. Ozawa et al. [52] reported that 830 nm diode irradiation at 3.8 J/cm^2^ on day 1 significantly increased the number of osteocalcin mRNA-positive cells and cell masses on day 2 and 4, respectively. Moreover, diode laser irradiation at 3 J/cm^2^ significantly enhanced osteocalcin synthesis in primary human osteoblast-like cells cultured on titanium implant material on day 10 [55]. Osteocalcin activity in hFOB 1.19 was significantly increased at 7 days after 940 nm laser irradiation at considerably high fluences in the range of 22.9–137.6 J/cm^2^ [42]. Even in hypoxic-cultured human osteoblasts, mRNA *Bglap* was increased on day 1–3 by diode laser irradiation at 1.2–3.6 J/cm^2^ [41]. The expression of *Bglap* on day 14 was significantly decreased in a human osteoblast cell line irradiated at 0.5, 1, and 2 J/cm^2^ [45]; however, indocyanine green (ICG)-mediated PBM significantly increased *Bglap* expression on day 7 following irradiation at 0.5 J/cm^2^ [46]. ICG-mediated PBM is a PBM with a photosensitizer, a light-activated molecule, and shares similar mechanisms with photodynamic therapy [46]. The effects of diode laser irradiation on osteoblasts have been investigated in the expression of type I collagen [31,34,35,40,41,45,46,47]. Most reports have shown that low-level irradiation at 0.5–3.6 J/cm^2^ significantly increased type I collagen expression in human osteoblastic cells at 1–20 days after irradiation [34,35,40,41,45,46,47]. Irradiation at higher fluences (5 and 15 J/cm^2^) also significantly increased *Col1a1* expression at 24, 48, and 72 h in a previous study [31]. Irradiation at 1.2–3.6 J/cm^2^ significantly increased the mRNA expression of type I collagen in hFOB 1.19 at 24 h after irradiation compared to that in hypoxic-cultured osteoblasts. However, at 48 and 72 h, type I collagen mRNA expression was significantly lower than that in hypoxic-cultured osteoblasts upon irradiation [41].

Several studies have reported the effect of diode laser irradiation on the expression of *Runx2*, an essential transcription factor for osteoblast differentiation [18,26,33,57,62,66]. Laser irradiation at 808 nm and 0.4 J/cm^2^ at continuous wave mode and 1.9 J/cm^2^ of 2 Hz pulsed mode at 830 nm significantly increased the expression of *Runx2* [18,33]. Ultrahigh-frequency and ultrashort-pulse 405 nm blue laser irradiation at 5.6 J/cm^2^ on osteoblasts significantly increased *Runx2* expression on day 3 in MC3T3-E1 cells [26]. Some reports showed that irradiation at 3 J/cm^2^ decreased *Runx2* expression in primary human osteoblast-like cells from alveolar bone [57,62].

Osterix is generally required for *Bglap* activation and bone formation [67] and is mutually regulated with Runx2 for the proliferation and differentiation of osteoblast-lineage cells and their progenitors [66]. Irradiation at 1.9–5.9 J/cm^2^ significantly increased the expression of *Osx* at 9 h on day 3 in osteoblasts [18,23,26,64]. In contrast, downregulation of *Osx* at 3, 6, and 12 h in primary human osteoblast-like cells from the alveolar bone after irradiation at 3 J/cm^2^ was reported [62].

BMPs, factors for bone formation, induce various genes, including *Runx2* and Osterix (*Sp7*) [68]. The effects of diode laser irradiation on BMP expression in osteoblasts have been studied previously [18,35,40,41,47,57]. The expression of *Bmp2*, *Bmp4*, and *Bmp7* was significantly increased at 6, 9, and 12 h after irradiation at 0.9–2.8 J/cm^2^ in MC3T3-E1 cells [18]. At later time points, on day 1–20, BMP mRNA expression was also significantly increased by irradiation at 1.2–6.7 J/cm^2^ [35,40,41]. Regarding bisphosphonate (BP)-related osteonecrosis of the jaw, a combined application of rhBMP-2 and irradiation at 1.2 J/cm^2^ was more effective in enhancing osteoblastic activity and bone formation activity in alendronate-treated hFOB 1.19 than the application of either modality alone [47].

BMPs belong to the TGF-β family, which is a prototype of a large family of cytokines involved in the growth and remodeling of bone [69]. TGF-β1 mRNA expression in osteoblasts was significantly increased at day 1–3, 10, and 20 after irradiation at 1.2–6.7 J/cm^2^ [35,40,41]. Laser irradiation at 830 nm and 3 J/cm^2^ significantly promoted TGF-β1 production, as measured by an enzyme-linked immunosorbent assay [55]. The expression of TGF-β1 suppressed by alendronate was recovered following a combined application of rhBMP-2 and irradiation at 1.2 J/cm^2^ in hFOB1.19 cells [47]. However, irradiation at 5–10 J/cm^2^ significantly decreased the expression of *TGFB1* in Saos-2 cells at 48 and 72 h [31].

Several previous studies have reported the expression of osteopontin [33,34,47,57]. Osteopontin, a bone matrix noncollagenous glycophosphoprotein, is secreted by osteoblasts during bone mineralization and remodeling [70]. Tani et al. [33] reported that red diode laser irradiation at 0.4 J/cm^2^ (λ = 635 nm) significantly increased osteopontin expression by densitometric analysis of the fluorescence intensity of immunostained osteopontin. Another study indicated that 808 nm laser irradiation at 1.2 J/cm^2^ had a greater effect on osteopontin expression in alendronate-treated hFOB 1.19 cells than in rhBMP-treated cells by Western blotting analysis [47]. However, 670 nm laser irradiation at 2 J/cm^2^ significantly decreased osteopontin mRNA expression in Saos-2 cells at 24 h compared to cells irradiated at 1 J/cm^2^ [34]. Irradiation at 3 J/cm^2^ also decreased osteopontin mRNA expression in primary human osteoblast-like cells at 14 days [57].

To investigate the effects of diode laser irradiation on bone remodeling, osteoclast-related markers (e.g., RANKL and OPG) have been studied [30,44,47,51,57]. A previous study reported a significant downregulation of RANKL, a significant upregulation of OPG, and a significant decrease in the RANKL/OPG ratio in primary rat calvarial cells irradiated at 1.1 J/cm^2^ [51], whereas irradiation with a dose of 3 J/cm^2^ increased the RANKL/OPG ratio in primary human osteoblast-like cells on titanium disks [57]. Irradiation at 5, 10, and 50 J/cm^2^ tended to increase the RANKL/OPG ratio, but no significant differences were observed [30]. Ga-Al-As laser irradiation with 808 or 920 nm at 1.2 J/cm^2^ increased RANKL and OPG expression in hFOB 1.19 cells [44,47].

Other factors related to bone formation are affected by diode laser irradiation. Smad1/5/8, which are activated by BMPs and referred to as BMP-specific receptor-regulated Smads [68,71], exhibited significantly enhanced phosphorylation by 805 nm laser irradiation at 5.9 J/cm^2^ [64]. The expression of phospho-extracellular signal-regulated kinase (ERK) was significantly increased by irradiation at 5 and 10 J/cm^2^ [36], and phosphorylated ERK1/2 was also increased at 15 min after irradiation at 2.9 J/cm^2^ [25]. The expression of distal-less homeobox 5 (Dlx5), which stimulates osteoblast differentiation [72], and Msh homeobox 2 (Msx2), which promotes osteoprogenitor proliferation but prevents differentiation, was significantly enhanced by irradiation at 1.9 J/cm^2^ in MC3T3-E1 cells [18]. Diode laser irradiation at 7.6 J/cm^2^ significantly increased osteoglycin gene expression at 2 h after irradiation [16]. Osteoglycin was reported to increase osteoblast differentiation in some studies [73], whereas other studies reported that osteoglycin decreases osteoblast differentiation [74].

### 3.2. Nd:YAG Laser

Nd:YAG laser is the second-most studied laser for its effect on osteoblasts or osteoblast-like cells after diode lasers. The effects of Nd:YAG laser on Saos-2 human osteoblast-like cells have been investigated in many studies [75,76,77,78,79]. Nd:YAG laser irradiation has been reported to have positive effects on osteoblasts or osteoblast-like cells. Irradiation at 10 Hz and 20 mJ for 10 s had a stimulatory effect on the cell viability and proliferation of Saos-2 cells at 7, 14, and 21 days [75]. Cell proliferation of Saos-2 was also significantly increased at 48 h following irradiation at 50 or 70 Hz and 20 mJ/pulse for 10 s [76]. Another study reported that irradiation 3 times at 0.5–2 W enhanced cell proliferation rates of Saos-2 cells on day 4 compared to one-time irradiation [77]. Irradiation at 10.3 J/cm^2^ accelerated cell migration until 24 h after irradiation and significantly enhanced ATP production in Saos-2 cells at 24 h following irradiation [78]. However, a Q-switched Nd:YAG laser irradiation at 1.5, 3, or 5 J/cm^2^ significantly decreased proliferation in MC3T3-E1 cells [80]. 

Nd: YAG laser irradiation showed various effects regarding gene and protein expression related to osteogenic differentiation or bone remodeling. ALP activity was significantly increased in MC3T3-E1 cells at 3, 7, and 14 days after irradiation at 1.5, 3, or 5 J/cm^2^, with or without nonglycosylated human recombinant BMP-2 (100 ng/mL) treatment [80]. ALP gene (*Alpl*) expression was significantly increased in Saos-2 cells at 24 h and 7 days after irradiation at 17.3 and 1.5 J/cm^2^, respectively [76,79]. An increase in Runx2 and osteopontin mRNA expression was also significantly induced at 7 days after irradiation at 1.5 J/cm^2^ in Saos-2 cells [76]. Expression of *BMP2* was significantly increased in MC3T3-E1 cells 2 days after irradiation at 3 J/cm^2^ [80]. Irradiation at 17.3 J/cm^2^ significantly increased mRNA expression of RANKL and OPG in Saos-2 cells at 24 h [79]. In a previous study, highly intensified calcium deposition on day 12 and significantly enhanced mineralization on day 21 were observed in MC3T3-E1 cells by Nd:YAG laser irradiation at 1.5–5 J/cm^2^ with or without rhBMP-2 treatment [80]. In addition, intracellular Ca^2+^ in Saos-2 cells was increased by irradiation at 50 Hz with a fluence of 1.5 J/cm^2^ through the activation of the transient receptor potential 1 (TRPC1) ion channels [76]. Gene expression of insulin-like growth factor-1 (IGF-1; *IGF1*), an important regulator of bone formation [81], was significantly enhanced in MC3T3-E1 cells on day 2 after irradiation at 3 J/cm^2^ with or without rhBMP-2 treatment [80].

### 3.3. Er:YAG Laser

Three previous studies reported the effects of Er:YAG laser irradiation on osteoblasts or osteoblast-like cells in vitro [82,83,84]. Er:YAG laser irradiation at 5.1–12.7 J/cm^2^ significantly reduced mitochondrial activity in Saos-2 cells compared to nonirradiated cells. However, mitochondrial activity was significantly increased with decreasing energy settings and/or increasing the distance between the laser application tip and the bottom of the culture plate [82]. Er:YAG laser irradiation at a fluence of 1.0–4.3 J/cm^2^ significantly increased MC3T3-E1 cell proliferation by irradiation in the absence of a culture medium. When irradiated at higher fluences (6.7 and 8.6 J/cm^2^), cell cytotoxicity of MC3T3-E1 was significantly increased. In the presence of a culture medium during irradiation, Er:YAG laser irradiation at much higher fluences (12.9 and 15.1 J/cm^2^) significantly increased MC3T3-E1 cell proliferation on day 1 and 3 without increasing cell cytotoxicity. The effect of Er:YAG laser on cell proliferation seemed to be induced by the activation of ERK [83], which plays a central role in the control of cell proliferation [85]. Another study reported that Er:YAG laser irradiation did not affect cell proliferation but significantly enhanced calcification of primary osteoblast-like cells from rat calvaria [84]. Irradiation at 3.3 J/cm^2^ significantly promoted mineralization of primary osteoblast-like cells on day 7, possibly via enhanced *Bglap* expression, without major thermal effects. Microarray analysis revealed that irradiation at 3.3 J/cm^2^ caused an upregulation of inflammation-related genes and downregulation of *Wisp2*, which plays an important role in the differentiation and mineralization of osteoblasts [86]. Gene set enrichment analysis showed that Er:YAG laser irradiation enriched Notch signaling, which plays a critical role in various cellular functions, including the promotion of osteogenic differentiation of osteoblasts in synergy with BMP [87]. 

### 3.4. Er,Cr:YSGG Laser

There are no reports on the direct effects of Er,Cr:YSGG laser irradiation on osteoblasts or osteoblast-like cells. Hence, the molecular biological effects of Er,Cr:YSGG laser irradiation on osteoblasts remains unclear.

### 3.5. CO_2_ Laser

The effect of CO_2_ laser irradiation on osteoblast-like cells was reported in a previous study [88]. The study investigated the effect of CO_2_ laser irradiation on rat osteoblast-like ROS 17/2.8 cells at 0.5–2 W for 20 s, resulting in a power density of 0.4–1.43 J/cm^2^. CO_2_ laser irradiation at 1.43 J/cm^2^ enhanced the mRNA expression of bone sialoprotein (BSP) at 12 h after irradiation. Transcription of BSP (*IBSP*) gene was also enhanced via the tyrosine kinase, Src tyrosine kinase, and ERK 1/2 signaling pathways, and fibroblast growth factor 2 response element in the rat *IBSP* gene promoter by CO_2_ laser irradiation.

### 3.6. Summary

The contents of this section are summarized in Table 1. Several reports have revealed the favorable effects of laser irradiation on osteoblasts or osteoblast-like cells. Laser irradiation enhances or increases cell proliferation, viability, migration, calcification, and expression of genes and proteins related to osteogenic differentiation, thereby promoting bone formation. These effects were observed in many studies using various types of lasers with different wavelengths; however, most of the effective energy fluences were low (under 6.0 J/cm^2^). In some studies, high power as well as low power irradiation was reported to have biological effects on osteoblasts or osteoblast-like cells. However, in vitro evidence related to Nd:YAG, CO_2_, Er:YAG, and Er,Cr:YSGG lasers are still limited. Further research is needed to elucidate the molecular biological effects of laser irradiation on osteoblasts.

## 4. Effects of Laser Irradiation on Fibroblasts

Fibroblasts are components of the connective tissue, which migrate to a lesion from the late inflammatory phase until epithelialization is completed [89]. Fibroblasts play an essential role in supporting other cells, are associated with wound healing or regeneration, and function to break down blood clots, thereby secreting various growth factors and cytokines and creating new extracellular matrix (ECM) and collagen structures [90]. Additionally, fibroblasts play a critical role in wound contraction [91]. Therefore, fibroblasts are essential for effective wound healing and tissue regeneration.

Since various types of lasers have been shown to enhance wound healing through tissue repair and anti-inflammatory effects in previous studies [92,93], the biological and molecular mechanism of this event has been pursued over the years. In particular, the effect of lasers on fibroblasts has been focused on this field. In this section, we focus on gingival fibroblasts.

### 4.1. Diode Laser

Diode lasers are representative lasers used in PBM for wound healing, and their biostimulatory effects, such as anti-inflammatory effects, have been reported in previous studies [4]. To determine the physiological mechanisms related to the biological effects, lipopolysaccharide (LPS)-challenged human gingival fibroblasts (HGFs) were irradiated using an 830 nm diode laser, with a total energy of 1.9–12.6 J corresponding to 3–20 min exposure, and PGE_2_ production and cyclooxygenase (COX)-1 (*COX1*) and COX-2 (*COX2*) gene expression were analyzed. The results suggested that PGE_2_ production and COX-2 mRNA levels were significantly suppressed in a dose-dependent manner upon laser exposure [94]. Additionally, dramatic downregulation of plasminogen activator (PA) activity, implicated in the degradation of extracellular matrix and synthesis of kinin in the process of inflammation, and downregulation of tissue PA mRNA levels were observed in the HGFs irradiated with 830 nm laser at 7.9 J/cm^2^ compared to that in the control group [95]. In addition, under similar conditions, interleukin (IL)-1β production was reduced, and further investigation by RT-PCR showed that mRNA expression of IL-1β was inhibited, whereas that of IL-1β-converting enzyme (ICE) was invariable [96].

The effect of diode laser on the proliferation and migration of fibroblasts has been reported previously [97,98,99], and the exposure time is more relevant to cell proliferation and cell survival than to power output [97]. Moreover, the cell proliferation rates in the single-dose and double-dose groups were compared using a 685 nm diode laser. Although cell proliferation was enhanced in both groups, no significant difference was observed between the two laser-irradiated groups. In addition, a single dose of 2.0 J/cm^2^ in the irradiated group resulted in a higher proliferation and viability rate than the nonirradiated control group. They also evaluated the secretion of growth factors such as basic fibroblast growth factor (bFGF), IGF-1, and the receptor of IGF-1 (IGFBP3). Single-dose irradiation significantly increased the secretion of bFGF and IGF-1 in irradiated cells, but the secretion of IGFBP3 was not significantly increased compared to that in control cells. All growth factors were significantly increased in the double-dose group compared to the nonirradiated group [100]. Similar to this result, the upregulation of mRNA expression for other growth factors such as IGF, VEGF, and TGF-β has also been confirmed [101,102,103]. In contrast, irradiation with an 810 nm laser caused a dramatic reduction in HGF cell numbers in vitro, with variable parameters, i.e., fluence of 24.6–492.8 J/cm^2^ [104].

Diode lasers have also been reported to have positive effects on collagen synthesis [102]. Investigation of the effect of a 904 nm diode laser on cell growth and procollagen synthesis of NIH-3T3 fibroblasts was approximately three- to sixfold after irradiation with 3 and 4 J/cm^2^, although no significant increase in procollagen synthesis was observed. However, neither cell growth nor procollagen synthesis was observed at 5 J/cm^2^ [105]. In contrast, in another study, gene expression of collagen type 1 was upregulated in the HGF cell line (HGF3-PI 53) 3 days after irradiation (4 J/cm^2^) [99].

To examine the laser’s effect on fibroblast-myofibroblast differentiation, NIH/3T3 fibroblasts were irradiated with a 635 nm diode laser at 0.3 J/cm^2^, and morphological, biochemical, and electrophysiological assays were conducted. Expression of matrix metalloproteinase (MMP)-2 and MMP-9 (MMPs play a pivotal role in physiological processes such as tissue remodeling) was upregulated, whereas tissue inhibitors of MMPs (TIMP)-1 and TIMP-2 were suppressed. Additionally, TGF-β1/Smad3‒mediated fibroblast-myoblast transition was inhibited. These results suggest that the diode laser modulates the TRPC1 ion channel, which in turn contributes to an antifibrotic effect by interfering with TGF-β1 signaling [106].

Bisphosphonate treatment is known to have a negative effect on wound healing [107]. In a study, diode laser irradiation tended to increase the viability of HGFs, although no significant difference was observed compared to that of nonirradiated control HGFs. However, when HGFs were cultured in a bisphosphonate-conditioned medium, laser irradiation significantly increased cell viability. Furthermore, laser irradiation on cell-free bisphosphonate-conditioned medium before culturing HGFs had no significant effect on cell viability, which indicated that laser irradiation directly affected the HGFs rather than suppressing the medicinal effect of bisphosphonate [43]. Thus, diode lasers may have the potential to become a supportive tool for preventing and treating of bisphosphonate-related diseases, such as osteonecrosis of the jaw.

However, adverse effects of lasers have also been reported. Diode laser irradiation (904 nm) at 3 J/cm^2^ on fibroblast cell line changed the ultrastructure of the cells’ cytoplasmic organelles; concurrently, a significant reduction in protein synthesis was observed [108]. Therefore, the cytotoxicity of the lasers should also be investigated for their safe usage.

### 4.2. Nd:YAG Laser

Previous studies have indicated that Nd:YAG lasers have various biological effects on cells, both in vivo and in vitro [4]. Nd:YAG laser (wavelength: 1060 nm) reduced collagen synthesis in human skin fibroblasts at energy levels as low as 1.1 × 10^3^ J/cm^2^, without altering DNA replication or cell viability [109]. Furthermore, DNA replication and collagen synthesis in human skin fibroblasts have been compared between Nd:YAG laser irradiation at 1.2–4.7 × 10^3^ J/cm^2^ for 3–12 s and under halogen lamp heat. Marked inhibition of DNA replication and collagen production was observed in the laser-irradiated fibroblasts, although no such decrease was noted in the halogen lamp-heated fibroblasts. Therefore, several characteristics other than the thermal effect may be critical in altering the biological functions of fibroblasts [110].

Nevertheless, histological analysis of laser-treated skin areas showed new collagen formation and increased the number of fibroblasts [111]. In addition, many other studies showed an increase in procollagen and collagen type-1 levels after Nd:YAG laser irradiation [76,112,113,114,115]. Nd:YAG laser downregulated the expression of MMP-1 and MMP-2 enzymes in the injured skin [112]. The reduction in MMP-1 was observed in keratinocyte-fibroblasts after Q-switched Nd:YAG laser irradiation at 8 J/cm^2^ [113]. Moreover, the effects of different wavelengths (532 nm and 1064 nm) of a Q-switched Nd:YAG laser on human skin fibroblasts were investigated. Both the lasers significantly increased the expression of type I and III procollagen and tissue inhibitors of metalloproteinase (TIMP)-1 and TIMP-2 and decreased MMP-2 and MMP-3 expression. Higher increased/decreased rates were observed in the 1064 nm Nd:YAG laser irradiation. Additionally, the 532 nm Nd:YAG laser increased Hsp70 and IL-6 expression, whereas the 1064 nm Nd:YAG laser upregulated TGF-β expression, suggesting that the molecular biological effects of Nd:YAG laser irradiation may differ according to the wavelengths used [115].

### 4.3. Er:YAG and Er,Cr:YSGG Lasers

The specific absorption characteristics of Er:YAG and Er,Cr:YSGG lasers have been reported to be beneficial for wound healing after soft tissue ablation [116,117]. During ablation, cells underlying the surface layer, including fibroblasts, indirectly receive low energy of the Er:YAG/Er,Cr:YSGG laser irradiation, which has been shown to promote wound healing and tissue regeneration [10]. Therefore, recently, a direct effect of low-level irradiation of Er:YAG/Er,Cr:YSGG laser on fibroblasts has been investigated in vitro. Pourzarandian et al. [118] showed that low-level laser therapy using an Er:YAG laser enhanced the proliferation of cultured HGFs and identified the optimal stimulative energy density of 3.4 J/cm^2^. They also observed a significant increase in PGE_2_ production and COX-2 mRNA expression after irradiation, and laser-induced PGE_2_ synthesis was completely inhibited by the COX-2 inhibitor, NS398 [119]. Proteomic analysis has been performed to investigate differentially expressed proteins in HGFs induced by low-level Er:YAG laser irradiation. On day 1 after irradiation at 2.1 J/cm^2^, significant cell proliferation without cell damage was observed. In addition, a total of 377 differentially expressed proteins were identified by mass spectrometry, 59 of which were upregulated and 15 were downregulated in laser-irradiated HGFs. Among the upregulated differentially expressed proteins, galectin-7, which is one of the essential proteins in the wound-healing process, was validated by quantitative PCR, Western blotting analysis, and enzyme-linked immunosorbent assay. To confirm the effect of galectin-7, HGFs were treated with recombinant human galectin-7, and cell proliferation was assessed in a dose-dependent manner, which suggested that alteration in protein expression and upregulation of galectin-7 may partly contribute to proliferation in HGFs [120]. Kong et al. [121] observed maximal cell proliferation at 6.3 J/cm^2^ on day 3 after irradiation, although it was accompanied by an increase in lactate dehydrogenase (LDH) release. An increase in ATP level, Ki-67 staining, and cyclin-A2 mRNA expression was confirmed, and it was observed that the increase in cell proliferation was due to the effect of Er:YAG laser irradiation on the cell cycle. However, alterations in the mitochondria and ribosomal endoplasmic reticulum (ER) were observed at 3 h postirradiation at 6.3 J/cm^2^; the changes subsided after 24 h, suggesting the occurrence of transient cellular injury. Furthermore, as the surface temperature of laser-irradiated cells reached 40.9 °C, nonirradiated cells were treated with a medium warmed at 40 °C, which also increased cell proliferation. In addition, laser-induced cell proliferation was suppressed by inhibitors of the thermosensory transient receptor potential channels (TRPV-1), capsazepine, or SKF96365. Finally, 21 genes involved in heat-related biological responses and endoplasmic reticulum-associated degradation were identified by microarray analysis. Therefore, 6.3 J/cm^2^ laser irradiation on HGFs may enhance cell proliferation through photothermal effects, despite transient cellular damage.

The effects of Er:YAG and Er,Cr:YSGG irradiation on cultured fibroblast cell lines (NCBI:C-165) were compared at different fluences: 1 W power output (10 Hz and 100 mJ) and 0.5 W power output (10 Hz and 150 mJ), respectively. Cell proliferation was upregulated in both groups compared to that in the control, but Er,Cr:YSGG laser irradiation tended to be more effective in cell proliferation than Er:YAG laser irradiation [122].

The mechanical effects of Er:YAG laser irradiation on fibroblasts have also been studied. As the number of primary human gingival fibroblasts significantly decreased after 3.0 W irradiation, gene expression analysis was conducted for cells irradiated at 0.6, 1.0, and 1.2 W. Cells were divided into four groups: control cells (not undergoing any procedures), cells undergoing only Er:YAG laser irradiation, cells undergoing only centrifugal loading, and a cells undergoing both Er:YAG laser irradiation and centrifugal force loading. Gene expression of *COX2*, *IL1B*, *TNFA*, *BMP2*, and *BMP4* was significantly increased in laser-irradiated cells (in a dose-dependent manner) compared to the control cells at 24 h after irradiation. Additionally, only *COX2* gene expression showed a significant increase in the centrifugal-loaded cells compared to control cells. In contrast, gene expression of *COX2*, *IL1B*, *TNFA*, *BMP2*, and *BMP4* was significantly higher in the laser-loaded and centrifugally loaded cells than in the centrifugally loaded cells. These results suggest that bone metabolism genes may be regulated by mechanical stimulation and laser irradiation combined [123].

### 4.4. CO_2_ Laser

High-power CO_2_ laser irradiation is mainly used in various surgical procedures as an alternative to traditional scalpel procedures [124]. Recently, low-level laser irradiation with a CO_2_ laser has gained attention in dentistry due to its promotive effect on wound healing [125,126,127]. The secretion of TGF-β1 was downregulated, whereas that of bFGF was upregulated by high-frequency CO_2_ laser irradiation, which occurred maximally at 4.7 J/cm^2^ in both normal and keloid dermal fibroblasts in vitro, resulting in enhancement of cell replication [128]. Thus, CO_2_ laser irradiation may have the ability to balance collagen organization in fibrosis. Furthermore, PBM with a CO_2_ laser on proliferation and migration was examined at the cellular level. Promotion of cell proliferation and migration of cultured human dermal fibroblasts (HDFs) were examined by MTS assay and cell migration assay, respectively, with irradiation of 1.0 J/cm^2^. In addition, with the same power, Western blotting analysis showed activation of Akt, ERK, and JNK signaling pathways. However, suppression of Akt, ERK, or JNK signaling pathways significantly inhibited both the proliferation and migration of laser-irradiated HDFs. The study indicated that low-level laser irradiation with a CO_2_ laser might promote proliferation and migration of fibroblasts via activation of Akt, ERK, or JNK signaling pathways [129].

From another perspective, as the clinical use of CO_2_ lasers has increased, the safety of laser irradiation has been investigated. Apfelberg et al. [130] exposed cultured fibroblasts to CO_2_ laser irradiation before the occurrence of malignancy was examined. The results showed that CO_2_ laser-irradiated cells did not exhibit a greater incidence of malignancy compared to controls, indicating that CO_2_ laser seems to be noncarcinogenic in laboratory cells.

### 4.5. Summary

The contents of this section are summarized in Table 2. The effects of PBM by laser irradiation on fibroblasts appear to be comparable despite the different wavelengths. Proliferation, migration, and secretion of cytokines/chemokines are the main functions affected by laser irradiation; which may lead to early wound healing; although, the biological/molecular evidence to support this phenomenon is still partial and inadequate. Moreover, excessive power or irradiation time results in cell damage and ineffective treatment. However, concurrently, lasers have the potential to regulate collagen synthesis through fibroblast stimulation depending on the target disease. Thus, further investigation on optimal configuration, which is consistent in vivo and in vitro, and more profound bioinformatic studies are required in the future to clarify the critical mechanisms of the effects of lasers on fibroblasts.

## 5. Effects of Laser Irradiation on Periodontal Ligament Cells

The periodontal ligament is the only ligament in the body that connects two distinct hard tissues. It is a fibrous, complex, and soft connective tissue that attaches the tooth root to the inner wall of the alveolar bone. The periodontal ligament thickness decreases with age. It is functionally essential for tooth support and for allowing teeth to withstand the forces generated during mastication [131].

### 5.1. Diode Laser

Diode laser irradiation has been reported to have positive effects on human periodontal ligament cells (hPDLCs). PBM at energy doses of 2 and 4 J/cm^2^ upregulated gene expression related to osteogenic differentiation, including *BMP2*, OC (*BGLAP*), *RUNX2*, and *ALPL* in hPDLCs. PBM enhanced the osteogenic differentiation of hPDLCs via cAMP regulation [132]. Additionally, it significantly increased cellular viability, decreased cellular inflammatory marker expression, and increased OC activity in hPDLCs at two energy densities (5 and 10 J/cm^2^) [133]. Suppression of inflammation is one of the positive effects of laser irradiation. After 670 nm Ga-Al-As laser irradiation (5 and 10 J/cm^2^), the mRNA expression of inducible NO synthase (*INOS*), *COX2*, and *IL1B* were decreased compared to that in nonirradiated control cells.

Another study using an 830 nm laser showed that laser irradiation at 3.8 J/cm^2^ decreased COX-2 and cytosolic phospholipase A_2_-α mRNA expression after 24 h in mechanically stretched hPDLCs [134]. The increase in PGE_2_ production was significantly inhibited by diode laser irradiation at 346–1152 J/cm^2^ in a dose-dependent manner. The increase in IL-1β production was also significantly inhibited by diode laser irradiation, although the inhibition was observed only with high-power irradiation [135]. Diode laser irradiation (4.0–7.9 J/cm^2^) significantly inhibited a marked increase in plasminogen activator activity in hPDLCs in response to stretching [136]. Additionally, Huang et al. [137] reported that the gene expression levels of *INOS*, *TNFA*, and *IL1B* in LPS-exposed periodontal ligament cells were decreased after irradiation, and phospho-ERK expression was significantly increased in the laser-irradiated cells compared to that in nonirradiated cells.

hPDLCs irradiated with an 810 nm diode laser, showed promotion of proliferation and differentiation. Irradiation at 3.9 J/cm^2^ increased proliferation of human periodontal ligament fibroblasts (PDLFs) between 24 and 48 h, and ALP activity at 48 and 72 h. The phosphorylated ERK level was also more prominent after irradiation at 3.9 J/cm^2^ energy fluency [138]. Additionally, the protein expression of MMP-8 in hPDLFs was decreased by 810 nm diode laser irradiation at 10 J/cm^2^ [139]. Moreover, the 809 nm diode laser irradiation at 2.0–7.8 J/cm^2^ of PDLFs significantly upregulated their proliferation up to 72 h [140].

### 5.2. Er:YAG Laser

Er:YAG laser irradiation at 4.2 J/cm^2^ on hPDLFs promoted cell proliferation, migration, and invasion abilities. The report also revealed that the silencing of galectin-7 abrogated the effects of Er:YAG laser on cell proliferation, migration, and invasion, suggesting that the Er:YAG laser promoted these effects through the induction of galectin-7 [141].

### 5.3. Nd:YAG Laser, Er,Cr:YSGG Laser, and CO_2_ Laser

There are no reports on the effects of Nd:YAG laser, Er,Cr:YSGG laser, and CO_2_ laser irradiation on periodontal ligament cells.

### 5.4. Summary

The contents of this section are summarized in Table 3. Laser irradiation on PDLs enhanced cell proliferation, migration, calcification, and differentiation. In addition, gene expression was altered by laser irradiation, especially with suppression of inflammatory products. However, the effects of laser irradiation on hPDLCs and hPDLFs were only investigated using diode and Er:YAG lasers. It is necessary to generate more evidence and reveal the mechanisms by which laser irradiation affects hPDLCs.

## 6. Effects of Laser Irradiation on Endothelial Cells

The mechanisms related to the effects of laser irradiation on wound healing are not completely clear. However, some studies reported that laser treatment could accelerate wound healing, especially in acute, chronic, and impaired wound-healing conditions [142] as well as in periodontal disease [143]. Endothelial cells play important roles in the process of wound healing and regeneration of periodontal tissue [144]. Hence, in this section, we summarized the direct effects of laser irradiation on endothelial cells.

### 6.1. Diode Laser

Some researchers have investigated the effect of diode laser irradiation on endothelial cells. The human vascular endothelial cell line (HECV) irradiated with an 808 nm diode laser (60 J/cm^2^) demonstrated no significant difference in viability but demonstrated higher proliferation than non-treated cells. Moreover, the study reported that diode laser stimulated mitochondrial oxygen consumption and ATP synthesis in HECV [145]. Another study using the 808 nm diode laser reported that CD54, CD62E, monocyte chemotactic protein-1 (MCP-1) expression, and von Willebrand factor release were altered in human umbilical vein endothelial cells (HUVECs) stimulated with IL-1β followed by laser irradiation. MCP-1 expression in HUVECs was significantly lower 6 h after 4.5 J/cm^2^ stimulation than in IL-1β stimulated cells. In addition, both 1.5 and 4.5 J/cm^2^ of laser irradiation inhibited IL-1β-induced increase in CD54 and CD62E concentration in the supernatant. Therefore, this study suggested that low-power laser irradiation decreased the pro-inflammatory and procoagulant activity of IL-1β-stimulated endothelial cells [146]. Moreover, a 670 nm diode laser irradiation caused a stimulatory effect on the proliferation of HUVECs [147] and increased their viability [43]. Using a 635 nm diode laser at different doses (2, 4, and 8 J/cm^2^), all doses of irradiation significantly increased the proliferation of HUVECs and significantly reduced the concentration of soluble vascular endothelial growth factor (sVEGFR-1), an inhibitor of vascular endothelial growth factor (VEGF), compared with nonirradiated cells [148]. A study using rhesus macaque choroid-retinal endothelial cells (RF/6A) reported that an 810 nm diode laser (over 84.0 J/cm^2^) irradiation caused significant cell death, and irradiation at a fluence of 45.9–76.4 J/cm^2^ induced Hsp70 hyperexpression at 12–18 h postirradiation [149].

### 6.2. Nd:YAG Laser

Some studies have reported the effect of Nd:YAG laser irradiation on endothelial cells. A significant induction in vinculin expression, a focal adhesion protein involved in cell adhesion and migration, in human endothelial H-end cells was observed in the Nd:YAG-irradiated (fluence, 1.5 J/cm^2^) cells. Moreover, this study showed that Nd:YAG laser irradiation did not affect cell viability and stimulated cell growth [76]. In another study, cultured rat aortic endothelial cells at 5 h after Nd:YAG (1.6 J/cm^2^) laser irradiation were examined using a DNA array chip. This study showed that 20 genes in laser-treated cells were upregulated by more than four-fold compared to those in the control, and Nd:YAG laser irradiation also upregulated gene expression related to cell migration, cell structure neurotransmission, and inflammation [150]. Moreover, Nd:YAG laser irradiation (1.5 J/cm^2^) of HUVECs cultured on titanium disks coated with *Porphyromonas gingivalis* LPS caused downregulation of endothelial adhesion molecules, including that of intercellular adhesion molecule-1 (ICAM-1) and vascular cell adhesion molecule (VCAM) levels compared to nonirradiated HUVECs [151].

### 6.3. Er:YAG, Er,Cr:YSGG, and CO_2_ Lasers

There are no reports on the direct effects of irradiation on endothelial cells. Therefore, the molecular biological effects of Er:YAG, Er,Cr:YSGG, and CO_2_ laser irradiation on endothelial cells are not apparent to date.

### 6.4. Summary

The contents of this section are summarized in Table 4. In conclusion, studies on the molecular biological effects on endothelial cells using lasers have not been so much reported and are inconclusive. Further research is needed to reveal the effects of diode lasers on endothelial cells.

## 7. Effects of Laser Irradiation on Cementoblasts

Cementum is a unique, avascular, and mineralized tissue formed by cementoblasts [152]. Only one study has reported the effects of laser irradiation on cementoblasts. Diode laser irradiation at 940 nm was performed on root plate- or microplate-seeded cementoblasts at a fluence of 18 J/cm^2^. Cell proliferation was not different until 96 h, but laser irradiation significantly retarded the decrease in cell proliferation after 96 h compared to the untreahted control group. Additionally, *Ibsp* and *Bglap*, which are transcripts required for cementum formation, were significantly increased in laser-irradiated cells compared to nonirradiated cells. Moreover, the expression levels of *Bmp-2,3,6,7* were significantly increased. These results indicate that biostimulation can be used during regenerative periodontal therapies to trigger cells with a periodontal attachment apparatus [153]. The contents of this section are summarized in Table 5.

## 8. Effects of Laser Irradiation on Epithelial Cells

Epithelial cells are found on the surfaces of tissues and organs. Although they share some common characteristics, they vary in size, shape, and general appearance, according to their location [154]. Moreover, they protect deeper tissues against the external environment and possess secretory and supportive functions, thus contributing to homeostasis maintenance. Epithelial cells play an important role in wound healing [155]. There are many studies on the effects of laser irradiation on epithelial cells associated with various tissues [156,157,158], with the exception of oral tissues. Herein, we summarize the effects of laser irradiation on oral epithelial cells. To our knowledge, no research has been published regarding the effects of Nd:YAG, Er:YAG, Er,Cr:YSGG, or CO_2_ lasers on oral epithelial cells. Although some studies have reported increased proliferation in Nd:YAG laser-irradiated epithelial cells [77], there are no studies on oral tissues.

### 8.1. Diode Laser

Diode laser irradiation has been previously reported to enhance wound healing [159]. Diode lasers can penetrate superficial tissues to exert their effects in deeper tissues. However, epithelial cells are the first cells that receive laser energy. Thus, epithelial cells absorb the highest amount of energy compared to other underlying cells. Therefore, the effects of diode lasers have been studied in vitro on various epithelial cells, including keratinocytes [160], a keratinocyte cell line (Hacat) [161,162], epithelial adenocarcinoma (HeLa) cells [163,164], pigment epithelial cells [165,166,167], and human breast epithelial cell lines (SVCT and Bre80hTERT) [168]. In this section, we focus on epithelial cells or cell lines related to oral tissues.

To examine the effect of a low level diode laser irradiation on oral epithelial cells, cultured, normal human oral keratinocyte (NOKSI) cells were irradiated using an 810 nm diode laser in continuous wave mode for 5 min, at a distance of 14.5 cm. Upregulated gene and protein expression of human β defensin-2 (HBD-2), a potent antimicrobial and wound-healing factor, were confirmed by qPCR, Western blotting, and immunostaining. Increased expression of HBD-2 was mediated by laser-activation of the TGF-β1 pathway [169].

A pulsed diode laser has also been used for in vitro studies on epithelial cells. In a study by Ejiri et al. [170], the effect of low-level diode laser irradiation was examined on primary human gingival epithelial cells (HGECs). Using a 904–910 nm diode laser applied at a high frequency of 30 kHz for 1–10 min, at 5.7–56.7 J/cm^2^, a significant increase in the proliferation of laser-irradiated cells was detected by WST-8 assay, 24 h after laser irradiation. The maximum proliferative effect was observed after 5 min of laser irradiation. The in vitro wound-healing assay showed a dramatic increase in migration among the laser-irradiated cells. Moreover, phosphorylation of MAPK/ERK was observed at 5, 15, 60, and 120 min after irradiation, whereas stress-activated protein kinases/c-Jun N-terminal kinase and p38 MAPK remained unphosphorylated. These results indicate that proliferation and migration of HGECs may be promoted via activation of MAPK/ERK.

The antimicrobial effects of diode lasers have been reported previously [171]. To investigate the cellular mechanisms of these effects, human oral squamous epithelial carcinoma cell lines (Ca9-22 and SCC-25) were treated with LPS. An 805 nm diode laser was then used to irradiate cells in a repeated pulse mode for 60 s at a 1 cm distance. The expression of *DEL1*, which encodes a protein with anti-inflammatory effects, was significantly increased following laser irradiation. In contrast, LPS-induced IL-6 and IL-8 expression was significantly suppressed following laser irradiation. A significant increase in migration was also observed in laser-irradiated cells [172]. These results indicate a suppressive effect of laser irradiation on the inflammatory response.

### 8.2. Summary

The contents of this section are summarized in Table 6. Many studies have examined the effect of laser irradiation on the epithelium in vivo [156,157,158]. However, despite the importance of epithelial cells in wound healing, in vitro studies are limited, especially with regard to epithelial cells associated with oral tissues. Epithelial cells have different characteristics based on their location and associated tissues, and further investigation of oral epithelial cells is required to enable the advancement of laser therapies for use in oral wound healing.

## 9. Effects of Laser Irradiation on Osteocytes

Only a few studies have reported the effects of laser irradiation on osteocytes. Suppression of *Sost* expression was observed in primary osteocyte-like cells isolated from rat calvaria after CO_2_ laser irradiation at 0.7–2.8 J/cm^2^ without an increase in temperature. The study reported an increase in *Dmp1* expression was after CO_2_ laser irradiation at 1.4–2.8 J/cm^2^ [173]. Ohsugi et al. [174] reported that *Sost* expression in osteogenic cells (osteoblast-like cells isolated from rat calvaria cultured with osteoinduction medium for 21 days) was decreased at 6 h after Er:YAG laser irradiation (2940 nm) at energy densities of 1.5 and 3.1 J/cm^2^. Following the results obtained by quantitative PCR, sclerostin (coded by *Sost*) expression in the cultured supernatant was significantly decreased. As sclerostin produced by osteocytes can inhibit osteoblast activity and suppress bone formation, it is thought that Er:YAG laser irradiation may promote bone formation via the suppression of *Sost* expression. The contents of this section are summarized in Table 7.

## 10. Effects of Laser Irradiation on Osteoclasts

Osteoclasts are multinucleated giant cells that have the capacity to resorb mineralized tissues [175]. The development of osteoclasts proceeds within the local microenvironment of the bone.

Only a single report has been published on the effect of laser irradiation on osteoclasts in vitro. Rat osteoclast precursor cells (osteoclast-like cells) purified from rat bone marrow were subjected to laser irradiation with 810 nm diode and a maximum power output of 50 mW at exposure times of 1, 3, 6, or 10 min/day, which corresponded to 9.3, 28.0, 56.0, or 93.3 J/cm^2^, respectively. Laser irradiation at 9.3–56.0 J/cm^2^ increased the number of tartrate-resistant, acid phosphatase-positive multinucleate cells. Furthermore, osteoclasts appeared on day 2 in the laser-irradiated groups but not until day 3 in the nonirradiated control groups. Receptor activator of NF-kappaB (RANK) in the laser-irradiated groups showed significantly greater staining compared to the control group on day 2 and 3 by immunohistochemistry, and the mRNA expression of RANK was upregulated, consistent with the immunohistochemistry results. The study suggested that irradiation with an 810 nm diode laser facilitated the differentiation and activation of osteoclasts via RANK expression [176]. The contents of this section are summarized in Table 8.

## 11. Effects of Laser Irradiation on Stem Cells

In multicellular organisms, stem cells are undifferentiated or partially differentiated cells that can differentiate into various types of cells and proliferate indefinitely to produce more numbers of the same stem cells. They are the earliest type of cells in the cell lineage [177]. Mesenchymal stem cells (MSCs) are multipotent cells found in adult tissues. Adult MSCs were isolated from almost every type of connective tissue, such as adipose [178], bone marrow, periodontal ligament [179], and dental pulp tissues [180]. Stem cell therapy is the use of stem cells to treat or prevent diseases. Stem cell therapy is applied to many types of treatments, including regeneration and wound healing [181]. Recently, the PBM effects of laser irradiation on MSCs have attracted much attention. In the next paragraph, we have reviewed the effect of laser irradiation on MSCs.

### 11.1. Diode Laser

Although studies on PBM’s effect on MSCs are limited, several studies using diode lasers with different wavelengths have been reported. PBM (using a combination of 630 and 810 nm lasers) stimulated the viability of human adipose-derived stem cells (hASCs) and human bone marrow mesenchymal stem cells (hBM-MSCs). In addition, PBM (irradiation once or twice at 630 nm and 0.6 and 1.2 J/cm^2^) increased the viability of hASCs compared to the control and laser-treated hBM-MSCs. Furthermore, PBM (using a combination of 630 and 810 nm lasers, 3 times irradiation at 2.4 J/cm^2^) increased hASC viability compared to control and laser-treated hBM-MSCs [182]. Some studies have reported the effect of laser irradiation at around 630 or 810 nm on MSCs. Diode laser irradiation at 635 nm in MSCs derived from femurs and tibias in rats caused an increase in the expression levels of v-akt murine thymoma viral oncogene homolog 1 (*Akt1*), cyclin D1 gene (*Ccnd1*), phosphatidylinositol 3-kinase, catalytic alpha polypeptide gene (*Pik3ca*), in addition to a decrease in protein tyrosine phosphatase nonreceptor type 6 (*Ptpn6*), and serine/threonine kinase 17b (*Stk17b*) expression. Microarray analysis was also performed in this study, which revealed that 119 genes were differentially expressed, and various genes involved in cell proliferation, apoptosis, and the cell cycle were affected. The study suggested that the increase in MSC proliferation was mediated through the PI3K/Akt/mTOR/eIF4E pathway [183]. In addition, cytotoxicity evaluated by LDH assay did not show a significant difference between nonirradiated and 635 nm diode laser-irradiated (0.5–5.0 J/cm^2^) MSCs obtained from rat bone marrow. Diode laser irradiation at 0.5 J/cm^2^ was found to be an optimal energy density to stimulate the proliferation of bone marrow stromal cells (BMSCs); additionally, irradiation at 5.0 J/cm^2^ significantly stimulated the secretion of VEGF and NGF. Furthermore, after 5-aza induction, myogenic differentiation was observed in all the groups, and diode laser irradiation at 5.0 J/cm^2^ dramatically facilitated the differentiation [184]. The effect of 635 nm diode laser irradiation on the osteogenic differentiation of MSCs has also been reported. Laser irradiation (0.4 J/cm^2^) on human mesenchymal stromal cells (hMSCs) increased vinculin-rich clusters, osteogenic expression of markers (e.g., Runx-2, alkaline phosphatase, osteopontin), and mineralized bone-like nodule structure deposition as well as induced stress fiber formation and upregulated the expression of the proliferation marker Ki67. The study suggested that 635 nm diode laser irradiation may be a potentially effective option for promoting/improving bone regeneration [33]. In addition, 635 nm diode laser irradiation (0.3 J/cm^2^) on MSCs derived from the femora and tibia of male C2F1 mice significantly enhanced MSC proliferation, without a change in cell viability. They also found that the increase in proliferation after 635 nm diode laser irradiation was associated with the upregulation and activation of the Notch-1 pathway and increased membrane conductance through voltage-gated K+, BK, and Kir channels and T- and L-type Ca^2+^ channels [185].

Recently, studies on diode laser irradiation at 808 nm related to MSCs have also been reported. At 0.5-4.0 J/cm^2^, irradiation of human gingival mesenchymal stem cells (HGMSCs) promoted their migration but not proliferation. Furthermore, diode laser irradiation could activate mitochondrial ROS, which could elevate the phosphorylation levels of JNK and IKB in HGMSCs, further activating NF-κB concomitantly with the elevation of the nuclear translocation of p65. Taken together, these results indicate that PBM may promote cell migration via the ROS/JNK/NF-κB pathway [186]. High power 808 nm diode laser irradiation (64 J/cm^2^) enhanced osteogenesis. Laser irradiation of BMSCs from 3-old female BALB/c mice increased the protein expression of Runx2 and Osterix and suppressed PPARγ, a pivotal transcription factor in adipogenic differentiation. Positive areas of ALP and Alizarin Red S histochemical staining were significantly increased after laser irradiation [187].

Regarding the 606 nm diode laser irradiation, irradiation with 1.9 J/cm^2^ enhanced the proliferation of BMSCs, although irradiation with 11.7 J/cm^2^ suppressed the proliferation. The cytotoxic effect of 50 µg/mL carboplatin was eliminated, and the inhibitory effect of 0.1 µg/mL vincristine was attenuated by laser irradiation at 1.9 J/cm^2^ [188]. In addition, 660 nm diode laser irradiation (5 J/cm^2^) on stem cells from human exfoliated deciduous teeth (SHEDs) increased cell proliferation and expression of mesenchymal stem cell markers, including OCT4, Nestin, and CD90 [189]. However, another study showed that human dental pulp stem cells (hDPSCs) irradiated at 660 nm and 5 J/cm^2^ showed signs of apoptosis and necrosis as observed by transmission electron microscopy (TEM). Diode laser irradiation at 3 J/cm^2^ increased fibronectin production in hDPSCs [190]. The effects of 660 nm diode laser irradiation (1.6 J/cm^2^) were also evaluated in hDPSCs. Gene expression of brain-derived neurotrophic factor (*BDNF*), glial cell line-derived neurotrophic factor (*GNDF*), matrix-associated protein 2 (*MAP2*), nuclear receptor-related 1 protein (*NURR1*), and dopamine transporter (*DAT*) were increased, especially in the first 7 days of dopaminergic induction. However, the hDPSCs were not able to differentiate into functional dopaminergic neurons either in nonirradiated control or laser-irradiated groups [191].

### 11.2. Nd:YAG Laser

There are some reports on the effects of Nd:YAG laser irradiation on MSCs. Nd:YAG laser irradiation (2 and 4 J/cm^2^) on hBMSCs promoted proliferation and osteogenesis, although irradiation at an energy density of 16 J/cm^2^ significantly suppressed the proliferation and osteogenesis of hBMSCs [192]. In addition, Nd:YAG laser irradiation at 9.8 J/cm^2^ on MSCs obtained from horses did not show a difference in viability between irradiated and control MSCs. However, laser-irradiated MSCs exhibited slightly lower proliferation and significantly increased expression of IL-10 and VEGF compared to nonirradiated control MSCs [193]. Frequency-doubled Nd:YAG laser irradiation (532 nm) of human adipose tissue-derived stem cells (hADSCs) was performed at densities of 5–45 J/cm^2^ for 30–300 s. Mitochondrial activity of hADSCs was evaluated by autofluorescence emission at wavelengths associated with nicotinamide adenine dinucleotide (NADH) and flavoproteins. Laser irradiation at 5–9.2 J/cm^2^ significantly increased the proliferation of hADSCs, which was attributed to an increase in mitochondrial activity, although hADSCs irradiated at 28 and 45 J/cm^2^ showed a significant decrease in proliferation and autofluorescence [194].

### 11.3. CO_2_ Laser

We found only one report that mentioned the effect of CO_2_ laser irradiation on MSCs in an in vitro study. CO_2_ laser irradiation (9 W, exposure time 4 ms/shot and a medium pattern of the spots) on hADSCs increased their proliferation when cultured under nutrient-deprived conditions (0.5% FBS) and reduced cell proliferation in a medium supplemented with 10% FBS. CO_2_ laser irradiation caused a transient increase in mitochondrial ROS and the capacity to restore Δψm after rotenone-induced depolarization, and increased the secretion of MMP-2 in conditioned media comprising MMP-9, VEGF, and adiponectin, which have the capacity to support the angiogenesis of endothelial progenitor cells. The study concluded that CO_2_ laser irradiation on ADSCs might activate the redox pathways that increase cell proliferation and enhance the secretion of angiogenic molecules [195].

### 11.4. Summary

The contents of this section are summarized in Table 9. Reports on the effects of laser irradiation on MSCs are limited. However, some studies have shown that laser irradiation, especially low-power irradiation, causes cell proliferation and favorable gene expression changes in MSCs. MSCs are already clinically applied for periodontal regeneration. MSC sheets transplanted to root surfaces can induce regeneration of periodontal tissue [196]. Although further research is required to clarify the effects of laser irradiation on MSCs, laser irradiation may enhance MSCs regenerative capabilities in periodontal tissues.

## 12. Conclusions

This review summarizes the effects of laser irradiation on cells related to periodontal tissues (Figure 1) and clearly shows that laser irradiation can have many positive effects on various cell types in periodontal tissues. Numerous studies have reported that laser irradiation enhances cell proliferation, migration, viability, calcification, gene expression, and protein expression. Additionally, the favorable effects on cells vary depending on the fluences and type of lasers. Irradiation using diodes or Nd:YAG lasers is clinically feasible and can be applied in association with any periodontal procedure, since it reaches deep tissues due to its deeply penetrating wavelength. By contrast, Er:YAG and CO_2_ lasers, which are only superficially absorbed, are only effective on epithelial cells and connective tissue surfaces, during nonsurgical periodontal treatments, or exposed bone and connective tissues, during periodontal surgeries. Although the purposes of periodontal treatment include anti-inflammation, tissue repair, and tissue regeneration, a single laser irradiation under a single specific irradiation condition cannot achieve all desired positive effects. Furthermore, a certain irradiation condition might have negative effects on some cells in periodontal tissues, since appropriate irradiation conditions vary with cell type. When applying a laser to regenerate periodontal tissues in the clinic, it is necessary to consider which cells need to be targeted and activated and then select the suitable laser type and energy fluence to match the cell type. However, there is still insufficient basic research on the PBM of lasers from this review. Because the type of lasers, irradiation time, distance, and fluence are quite varied, it is difficult to critically determine the optimal criteria for laser usage. From this review, we realize the promising PBM effects of lasers in periodontal therapy, and we can gain insights regarding the appropriate fluences to be utilized in laser applications to osteoblasts, fibroblasts, and MSCs. We will continue to research lasers for periodontal phototherapy, including regeneration of periodontal tissues in the future.

## Figures and Tables

**Figure 1 ijms-21-09002-f001:**
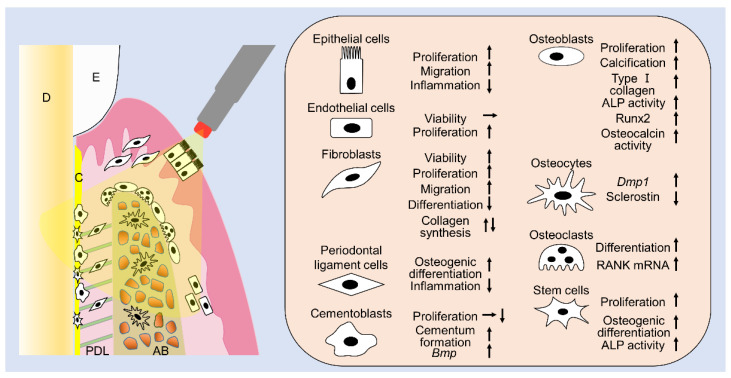
The summary of this review. Laser irradiation has various effects on cells related to periodontal tissues. E: enamel, D: dentin, PDL: periodontal ligament, AB: alveolar bone.

**Table 1 ijms-21-09002-t001:** Summary of the effects of laser irradiation on osteoblasts.

Reference No.	Laser	Cell	Year Author	Irradiation Protocol	Major Findings
[15]	Diode	MC3T3-E1	2001 Yamamoto, et al.	830 nm 7.64 J/cm^2^ CW 20 min	Irradiation may enhance DNA replication and play a role in stimulating proliferation of osteoblast through the enhancement of the mouse minichromosome maintenance 3 gene expression.
[16]	Diode	MC3T3-E1	2003 Hamajima et al.	830 nm 7.64 J/cm^2^ CW 20 min	The osteoglycin gene was upregulated at 2 h after low level laser irradiation.
[17]	Diode	MC3T3-E1, MG-63	2007 Renno, et al.	670, 780, or 830 nm 0.5, 1, 5, or 10 J/cm^2^ CW	Osteoblast proliferation increased significantly after 830 nm laser irradiation (10 J/cm^2^) but decreased after 780 nm laser irradiation (at 1, 5, and 10 J/cm^2^). MG-63 cell proliferation increased significantly after 670 nm (at 5 J/cm^2^) and 780 nm (at 1, 5, and 10 J/cm^2^) laser irradiation, but not after 830 nm laser irradiation. Alkaline phosphatase (ALP) activity in the osteoblast line was increased after 830 nm laser irradiation at 10 J/cm^2^
[18]	Diode	MC3T3-E1	2010 Fujimoto et al.	830 nm 0.97, 1.91, or 3.82 J/cm^2^ 2 Hz 5, 10, or 20 min	Expression of bone morphogenetic protein (BMP)-2, 4, and 7 were significantly increased at 6, 9, 12 h. Runt-related protein transcription factor 2 (*Runx2*), *Osx*, distal-less homeobox 5 (*Dlx5*), and Msh homeobox 2 (*Msx2*) expression was significantly increased at 12, 24, 48 h.
[19]	Diode	MC3T3-E1	2011 Kanenari, et al.	830 nm 7.64 J/cm^2^ 20 min	Laser irradiation enhances *Map1a* gene expression and modulates microtubule assembly and the functional structure of microtubules, in turn, stimulates osteoblastic proliferation and differentiation.
[20]	Diode	MC3T3-E1	2014 Migliario, et al.	980 nm 1.57, 7.87, 15.74, or 78.75 J/cm^2^ CW 1, 5, 10, 25, or 50 s	Laser irradiation enhances cell proliferation via ROS production.
[21]	Diode	MC3T3-E1	2014 Pagin, et al.	660 and 780 nm 3, 5 J/cm^2^ Punctual irradiation mode 2 and 5 s	Laser irradiation significantly promoted cell growth at 24 h.
[22]	Diode	MC3T3-E1	2017 Oliveira, et al.	660 or 780 nm 1.9, or 3.8 J/cm^2^ CW 4 or 8 s	Laser irradiation at both wavelengths significantly increased cell viability on 24 and 48 h. Infrared Ga-Al-As laser at 780 nm significantly increased ALP activity on 24 and 72 h. Red laser at 660 nm significantly increased matrix metalloproteinase (MMP)-2 activities on 48 and 72 h.
[23]	Diode	MC3T3-E1	2017 Son, et al.	808 nm ± 5 nm 1.2 J/cm^2^ CW15 s 3 times at 0, 24, and 48 h	Laser irradiation with melatonin treatment increased significantly Osterix (*Sp7*) expression at 48 and 72 h and ALP activity and calcification on day 7 and 14.
[24]	Diode	MC3T3-E1	2017 Li, et al.	808 nm 1.25, 3.75, or 6.25 J/cm^2^ CW 30, 90, or 150 s	Irradiation at 3.75 J/cm^2^ increased the cell amount at S phase and promoted cell proliferation through hedgehog signaling pathway at 24 h. The expressions of *Ihh*, *Ptch*, *Smo*, and *Gli* were significantly increased by 3.75 J/cm^2^ irradiation.
[25]	Diode	MC3T3-E1	2018 Kunimatsu, et al.	910 nm 0, 1.42, 2.85, 5.7, or 17.1 J/cm^2^ Pulsed 30 kHz	Cell proliferation was significantly increased by laser irradiation at a dose of 2.85, 5.7, or 17.1 J/cm^2^. Laser irradiation at a dose of 2.85 J/cm^2^ induced MC3T3-E1 cells to migrate more rapidly than nonirradiated control cells. Irradiation with the high-frequency 910 nm diode laser at a dose of 2.85 J/cm^2^ induces phosphorylation of Mitogen‑activated protein kinase (MAPK)/extracellular signal-regulated kinase (ERK)1/2 at 15 and 30 min later irradiation
[26]	Diode	MC3T3-E1	2018 Mikami, et al.	405 nm 1.9, 5.6, 9.4, 13.1, or 16.9 J/cm^2^ Pulsed 80 MHz 1 min	Laser irradiation significantly accelerated cell proliferation activity on day 3 and ALP activity on day 7 via transient receptor potential vanilloid 1(TRPV1). Expression of *Alpl*, *Sp7*, and *Runx2* mRNAs was significantly increased. Calcification was significantly increased 3 weeks later after final irradiation on day 2, 4, 6, 9, and 11.
[27]	Diode	Saos-2	2000 Coombe, et al.	830 nm 1.7–25.1 J/cm^2^ CW Single or multiple for 10 days	Cellular proliferation or activation of osteoblastic cells was not significantly affected by laser irradiation.
[28]	Diode	Saos-2	2013 Bayram, et al.	808 nm 1.316, or 2.63 J/cm^2^ CW 10 s	Laser irradiation lessened the detrimental effects of zoledronate, improved cell function and/or proliferation, and ALP activity.
[29]	Diode	Saos-2	2013 Bloise, et al.	659 nm 1 and 3 J/cm^2^ 200 or 600 s Single or multiple for 3 days	Cell proliferation is significantly increased on day 2 by single dose of 1 J/cm^2^ and on day 2, 3, and 7 with multiple doses of 1 and 3 J/cm^2^. ALP activity on day 14 and calcification on day 14 were increased significantly.
[30]	Diode	Saos-2	2014 Incerti Parenti, et al.	915 nm 1, 5, 10, 20, or 50 J/cm^2^ Pulsed 100 Hz 10, 48, 96, 193, or 482 s	Cell viability was significantly increased on day 3 by irradiation at 10 J/cm^2^, but significantly decreased by irradiation at 20 and 50 J/cm^2^. A rapid and transitory trend toward increased receptor activator of NF-κB ligand (RANKL)/osteoprotegerin ( OPG) ratio and a tendency toward a delayed increase in VEGF release for doses of 1 to 10 J/cm^2^ was found.
[31]	Diode	Saos-2	2015 Tschon, et al.	915 nm 5, 10, or 15 J/cm^2^ Pulsed 100 Hz 0, 48, 96, or 144 s	Wound healing was significantly promoted at 72 and 96 h. *COL1A1* expression was significantly increased at 24, 48, and 72 h. transforming growth factor (*TGF*) *B1* expression significantly decreased *TGFB1* expression at 48 and 72 h.
[32]	Diode	HOB and Saos-2	2016 Heymann, et al.	670 nm 100 mW/cm^2^ CW 120 s	Laser irradiation alone increased cell bioavailability.
[33]	Diode	Saos-2	2018 Tani, et al.	635 or 808 nm 0.378 J/cm^2^ CW 30 s	Laser irradiation caused no differences in viability at 24 h. Laser irradiation increased expression of *RUNX2*, *ALPL*, and osteopontin (*SPP1*) on 7 days and calcification on 18 days by activation of Akt signaling.
[34]	Diode	Saos-2	2008 Stein, et al.	670 nm 1 or 2 J/cm^2^ CW 30 s or 1 min	Cell viability, alkaline phosphatase activity, and the expression of osteopontin and collagen type I mRNA were slightly enhanced in cells irradiated with 1 J/cm^2^.
[35]	Diode	MG-63	2009 Saracino, et al.	910 nm 6.7 J/cm^2^ 30 kHz 5 min	Laser irradiation decreased cell growth, induced expression of *TGFB2*, *BMP-4*, and *BMP-7*, type I collagen, *ALPL*, and osteocalcin, and increased the size and the number of calcium deposits.
[36]	Diode	MG-63	2012 Huang, et al.	920 nm 5, or 10 J/cm^2^ 50–60 Hz 2.5 or 5 s	Laser irradiation promoted cell adhesion at 12 h and cell viability at 1 and 12 h. Laser irradiation reduced the expression of the lipopolysaccharide (LPS)-induced inflammatory markers iNOS (*INOS*), tumour necrosis factor α (*TNFA*), and *IL1B* and increased the expression of phospho-ERK.
[37]	Diode	MG-63	2014 Huertas, et al.	940 nm 0.5, 1, 1.5, or 2 W/cm^2^ 70 mW 1, 2, 3, 4, or 5 J	At 24 h culture, cell proliferation was increased.
[38]	Diode	MG-63	2013 Incerti Parenti, et al.	915 nm 2 J/cm^2^ CW 17, 31, or 157 s	Laser irradiation did not interfere in cell viability and proliferation.
[39]	Diode	MG-63	2014 Medina-Huertas, et al.	940 nm 1, 1.5 W/cm^2^ 3, 4 J 10.6, 12.96, 14.7, or 19.31 s	ALP activity was increased significantly at 24 h by irradiation at 1 W/cm^2^ and 3 J. CD54, CD86, and HLA-DR were decreased at 24 h.
[40]	Diode	MG-63	2015 Manzano-Moreno, et al.	940 nm 1, or 1.5 W 3 or 4 J	Laser irradiation significantly increased gene expression of *RUNX2*, *SP7*, *COL1A*, *ALPL*, *BMP2*, and *TGFB1* at 24 h.
[41]	Diode	hFOB 1.19	2013 Pyo, et al.	808 ± 3 nm 1.2, 2.4, or 3.6 J/cm^2^ CW 15 s at 0, 24, 48 h	Laser irradiation on hypoxic-cultured osteoblast stimulates osteoblast differentiation and proliferation at 24 and 72 h. Laser irradiation significantly increased expression of *BMP-2*, osteocalcin, Type 1 collagen, and *TGFB1*. Type 1 collagen expression were significantly decreased at 48 and 72 h.
[42]	Diode	hFOB 1.19	2013 Jawad, et al.	940 nm 22.92, 45.85, 68.78, 91.79, or 137.57 J/cm^2^ CW 3 or 6 min/day for 7 days	Laser irradiation significantly increased proliferation and ALP activity on day 3 and 7. Osteocalcin activity was significantly increased on day 7.
[43]	Diode	HHOB-c; Human osteogenic cells	2015 Walter, et al.	670 nm 280 mW 60 s CW	Laser irradiation increased the viability of cells, but was significant only in the experimental approach with pamidronate.
[44]	Diode	hFOB 1.19	2016 Shin, et al.	808 ± 3 nm 1.2 J/cm^2^ CW 15 s 3 times at 1, 24, 48 h	Laser irradiation significantly increased cell viability at 72 h. Expression of *RANKL* and M-CSF (*CSF1*) were significantly increased at 72 h.
[45]	Diode	Human osteoblasts cell line (ATCC^®^ CRL-11372)	2017 Bolukbasi Ateş, et al.	635 or 809 nm 0.5, 1, or 2 J/cm^2^ CW 10, 20, or 40 s	Viability was significantly increased at 48 and 72 h. Expression of *COL1A* was significantly increased by 1 and 2 J/cm^2^ at day 14. Expression of *BGLAP* was significantly decreased by 0.5, 1 and 2 J/cm^2^ at day 14.
[46]	Diode	Human osteoblasts cell line (ATCC^®^ CRL-11372)	2018 Ateş, et al.	809 nm 0.5, 1, or 2 J/cm^2^ CW 10, 20, or 40 s	Laser irradiation at 2 J/cm^2^ significantly increased cell viability at 24 h. ALP activity was significantly enhanced on day 7. Mineralization was significantly increased on day 14. The expression of *ALPL*, *COL1A*, and *BGLAP* was significantly increased on day 7 and/or 14.
[47]	Diode	hFOB 1.19	2018 Jeong, et al.	808 ± 3 nm 1.2 J/cm^2^ CW 15 s	Combined application of rhBMP-2 and laser irradiation was more effective than application of either modality alone. Expression of *RANKL*, *OPG*, and *M-CSF* in hFOB cells were increased following application of rhBMP-2 and laser irradiation. The expression of *TGFB1*, *BMP2*, collagen type I, and osteopontin were increased following combined application of rhBMP-2 and laser irradiation.
[48]	Diode	Osteo-1; Rat calvarial osteoblast-like cells	2006 Fujihara, et al.	780 nm 3 J/cm^2^ CW 12 s	Irradiation significantly increased cell proliferation with and without dexamethasone.
[49]	Diode	Rat primary calvarial osteoblastic cells	2006 Fukuhara, et al.	905 nm 1.25, 3.75, or 6.25 J/cm^2^ 150, 450, or 750 s Every day for 1–3 weeks	Irradiation energy of 3.75 J/cm^2^ induced an increased number of cells at day 3 and the greatest bone formation at day 21. Low-energy laser irradiation increased *Runx2* expression and ALP-positive colonies. FACS data demonstrated a higher proportion of cells in the G2/M phase of the cell cycle 12 h after irradiation compared with the control.
[50]	Diode	Rat primary calvarial osteoblastic cells	2007 Shimizu, et al.	830 nm 3.82 J/cm^2^ CW 10 min	Irradiation increased bone nodule formation at day 24 post-irradiation which is partly mediated by insulin-like growth factor-1 (IGF-I) expression.
[51]	Diode	Rat primary calvarial osteoblastic cells	2009, Xu, et al.	650 nm 1.14 or 2.28 J/cm^2^ 6000 Hz 5 or 10 min	Laser irradiation may directly promote osteoblast proliferation on day 3 and differentiation. Irradiation significantly downregulated RANKL and upregulated OPG, downregulating the RANKL:OPG mRNA ratio in osteoblasts.
[52]	Diode	Rat primary calvarial osteoblastic cells	1998 Ozawa, et al.	830 nm 3.82 J/cm^2^ CW 10 min	Laser irradiation significantly stimulated cellular proliferation, ALP activity, and osteocalcin gene expression thereafter. Laser irradiation at earlier stages of culture significantly stimulated bone nodules formation in the culture dish on day 21.
[53]	Diode	Rat primary calvarial osteoblastic cells	2001 Ueda and Shimizu	830 nm 0.48–3.84 J/cm^2^ CW or pulsed (1,2, 8 Hz) 1.25–10 for 2.5–20 min	Both CW and pulsed irradiation significantly enhanced cell proliferation, bone nodule formation, ALP activity, and *Alpl* gene expression as compared with the nonirradiated group.
[54]	Diode	Rat primary calvarial osteoblastic cells	2003 Ueda and Shimizu	830 nm 0.48–3.84 J/cm^2^ CW or pulsed (1, 2, and 8 Hz) 1.25–10 for 2.5–20 min	Laser irradiation on day 1 at all conditions significantly stimulated cellular proliferation on day 6, 9, and 12 as compared with the controls.
[55]	Diode	Primary human osteoblast-like cells from mandibular	2005 Khadra, et al.	830 nm 1.5 and 3 J/cm^2^ CW For 3 consecutive days	Greater cell proliferation in the irradiated groups was observed first after 96 h. Osteocalcin synthesis and TGF-b1 production were significantly greater on the samples exposed to 3 J/cm^2^
[56]	Diode	Rat primary calvarial osteoblastic cells	2020 Cardoso, et al.	660 or 808 nm 5, 8.3 J/cm^2^ CW 3 and 5 s	Laser irradiation at both wavelengths promoted cell proliferation and wound healing. ALP activity and mineralization were significantly increased.
[57]	Diode	Primary human osteoblast-like cells from alveolar	2010 Petri, et al.	780 nm 3 J/cm^2^ CW 9 min on day 3 and 7	Laser irradiation did not influence culture growth, ALP activity, and mineralized matrix formation. Gene expression of *ALPL*, *BGLAP*, *IBSP*, and *BMP7* was higher in laser-treated cultures, while *RUNX2*, *SPP1*, and OPG (*TNFRSF11B*) were lower on day 14.
[58]	Diode	Rat primary calvarial osteoblast-like cells	2013 Emes, et al.	808 nm 1.5 J/cm^2^ CW 90 s	Laser irradiation did not affect cell proliferation and ALP activity.
[59]	Diode	Primary human osteoblast-like cells from femur	2019 Morsoleto, et al.	808 nm 2 J/cm^2^ 5 s Every day for 8 days	Laser irradiations on 1–8 days enhanced cell viability and matrix mineralization on day 18.
[60]	Diode	A mouse OFCOL Ⅱ cell line	2008 Pires Oliveira, et al.	830 nm 3 J/cm^2^ CW 36 s	Cellular viability was significantly increased at 24, 48, and 72 h after irradiation. Intense grouping of mitochondria in the perinuclear region was observed at 24 and 48 h. Changes from a filamentous to a granular appearance in mitochondrial morphology and mitochondria distributed throughout the cytoplasm were observed 72 h.
[61]	Diode	Osteoblasts from rat bone marrow stem cells	2000 Dortbudak, et al.	690 nm 1.6 J/cm^2^ CW 60 s 3 times on day 3, 5, and 7	Irradiations 3 times on day 3, 5, and 7 significantly enhanced more fluorescent bone deposits than the nonirradiated cultures.
[62]	Diode	Primary human osteoblast-like cells from alveolar	2011 Grassi, et al.	920 nm 3 J/cm^2^ CW 60 s	Laser irradiation significantly enhanced Alp activity on day 7 and 14 and mineralization after 5 weeks. Laser irradiation decreased *Runx2* and *Sp7* mRNA at 3, 6, and 12 h.
[63]	Diode	Primary human osteoblast-like cells from mandibular	2018 Mergoni, et al.	915 nm 5, 15, and 45 J/cm^2^ CW 4, 12, 36, 41.7, 125, and 375 s Every day for 3 or 6 days	Irradiation for 6 days significantly increased bone deposits 3 weeks after irradiation.
[64]	Diode	Mouse primary calvarial osteoblasts	2010 Hirata, et al.	805 nm 2–12 J/cm^2^ CW 2 min	Irradiation stimulated BMP2-induced phosphorylation of Smad1/5/8 and *Bmp2* expression, but had no effect on the expression of inhibitory *Smads6*, *Smad7*, *Bmp4*, or insulin-like growth factor 1. Laser irradiation enhanced Smad-induced Id1 reporter activity and BMP-induced transcription factors such as Id1, Osterix, and Runx2. Laser irradiation also stimulated BMP-induced expressions of type I collagen, osteonectin, and osteocalcin mRNA.
[75]	Nd:YAG	Saos-2	2006 Arisu, et al.	1064 nm 20, 60, 80, and 120 mJ 0.2, 0.6, 0.8, 0.9, 1.2, 1.6, 1.8, 2.4, and 3.6 W Pulsed 10, 15, 20, and 30 Hz 10 s	Irradiation had a stimulatory effect on the cell viability and proliferation at 7, 14, and 21 days.
[76]	Nd:YAG	Saos-2	2010 Chellini, et al.	1064 nm 1.5 J/cm^2^ Pulsed 50 and 70 Hz 1.4 W 20 mJ 10 s	Laser irradiation did not affect cell viability but significant increased proliferation at 48 h. Laser irradiation significantly induced the expression of *ALPL*, *RUNX2*, and *SPP1* on day 7. Laser irradiation increased the intracellular Ca^2+^ levels through the activation of transient receptor potential 1 (TRPC1) ion channels.
[77]	Nd:YAG	Saos-2	2018 Kara, et al.	1064 nm Pulsed 5, 10, 20, and 30 Hz 0.5, 1, 2, and 3 W 100 mJ 30 s	The proliferation rates on day 4 increased as the number of applications increased, especially in those cases in which the irradiation was performed 2 or 3 times more.
[78]	Nd:YAG	Saos-2	2019 Tsuka, et al.	1064 nm 10.34 J/cm^2^ 10 pps 0.3 W 30 mJ 60 s	Laser irradiation accelerated migration of cells until 24 h, significant enhancement of ATP production.
[79]	Nd:YAG	Saos-2	2020 Tsuka, et al.	1064 nm 5.17, 17.23, 34.47, and 51.7 J/cm^2^ Pulsed 20–30 Hz 0.6, 2.0, 4.0, and 6.0 W 15 s	Laser irradiation significantly increased expression of *ALPL*, *RANKL*, *TNFRSF11B*, and RANKL/OPG ratio at 24 h.
[80]	Nd:YAG	MC3T3-E1	2010 Kim, et al.	1064 nm 1.5, 3, and 5 J/cm^2^ 15 pps 0.75 W 4–12 s	Laser irradiation significantly decreased cell proliferation at day 3, but significantly increased ALP activity on day 3, 7, and 14. Laser irradiation highly intensified calcium deposition at all fluences on day 12 and significantly enhanced mineralization on day 21. Laser irradiation significantly increased expression of *Bmp2*, *Cbfa1*, *SP7*, *Dlx5*, *Igf1*, and *Vegf*.
[82]	Er:YAG	Saos-2	2004 Schwarz, et al.	2940 nm 5.08, 7.62, 10.16, and 12.7 J/cm^2^ Pulsed 10 Hz 40, 60, 80, and 100 mJ 10 s	Mitochondrial activity increased significantly with decreasing energy settings and increasing distances.
[83]	Er:YAG	MC3T3-E1	2010 Aleksic, et al.	2940 nm 0.7, 1.0, 1.4, 2.1, 2.9, 3.1, 3.6, 4.3, 4.7, 6.4, 6.7, 8.6, 10.8, 12.9, 15.1, and 17.2 J/cm^2^ Pulsed 10, 20, 30, 40, and 50 Hz 23, 39, 50, and 68 mJ 30, 60, 90, and 120 s and 2.5, 3, 3.5, and 4 min	Significantly higher proliferation was also observed in laser-irradiated MC3T3-E1 cells at a fluence of approximately 1.0–15.1 J/cm^2^, whereas no increase in lactate dehydrogenase (LDH) activity was observed. Low-level Er:YAG irradiation induced phosphorylation of extracellular signal-regulated protein kinase (MAPK/ERK) 5–30 min after irradiation.
[84]	Er:YAG	Rat primary calvarial osteoblast-like cell, MC3T3-E1	2020 Niimi, et al.	2940 nm 2.2, 3.3, and 4.3 J/cm^2^ Pulsed 20 Hz 17.6, 26.4, and 34.5 mJ 60 s	Calcification and *Bglap* expression were significantly increased after Er:YAG laser irradiation at 3.3 J/cm^2^. Laser irradiation at 3.3 J/cm^2^ caused upregulation of inflammation-related genes, downregulation of *Wisp2*, and enrichment of inflammation-related and Notch signaling gene sets.
[88]	CO_2_	Rat osteoblast-like ROS17/2.8 cells	2011 Sasaki, et al.	0.357, 0.715, 1.07, and 1.43 J/cm^2^ 0.5, 1, 1.5, and 2 W 20 s	*Ibsp* mRNA levels were increased at 12 h after irradiation at /.1.43 J/cm^2^.

**Table 2 ijms-21-09002-t002:** Summary of the effects of laser irradiation on fibroblasts.

Reference No.	Laser	Cell	Year Author	Irradiation Protocol	Major Finding
[43]	Diode	Primary human gingival fibroblast cells (HGF cells)	2015 Walter et al.	670 nm 280 mW CW 60 s	Laser irradiation nonsignificantly increased cell viability compared to nonirradiated control HGFs. In bisphosphonate-treated HGFs, laser irradiation significantly increased cell viability compared to the control.
[94]	Diode	Lipopolysaccharide (LPS)-challenged human gingival fibroblast cells (HGF cells)	2000 Sakurai et al.	830 nm 0.95–6.32 J/cm^2^ CW 3–20 min	Laser irradiation suppressed LPS-induced PGE_2_ production by reducing cyclooxygenase (COX)-2 mRNA expression.
[95]	Diode	LPS-challenged human gingival fibroblast cells (HGF cells)	2000 Takema et al.	830 nm 7.90 J/cm^2^ CW 10 min	Plasminogen activator activity was dramatically elevated by LPS in cultured medium of HGF cells, which was significantly inhibited by laser irradiation in a dose-dependent manner.
[96]	Diode	LPS-challenged human gingival fibroblast cells (HGF cells)	2001 Nomura et al.	830 nm 3.95–7.90 J/cm^2^ CW 10 min	Interleukin (IL)-1β was dramatically elevated by LPS in cultured medium of HGF cells, which was significantly inhibited by laser irradiation in a dose-dependent manner.
[97]	Diode	Human gingival fibroblast cell line (LMF)	2001 Almeida-Lopez et al.	670, 780, 692, or 782 nm 2.0 J/cm^2^ CW	Laser irradiation promoted cell proliferation in vitro. Shorter irradiation time resulted in higher proliferation.
[98]	Diode	Continuous cell line; Ethics Committee 64/99-Piracicaba Dental School	2012 Basso et al.	730 ± 3 nm 0.5, 1.5, 3.0, 5.0, or 7.0 J/cm^2^ CW 40, 120, 240, 400, or 560 s	0.5 and 3.0 J/cm^2^ laser irradiation significantly increased cell metabolism, cell number, and cell migration.
[99]	Diode	Human gingival fibroblast cell line (HGF3-PI 53)	2013 Frozanfar et al.	810 nm 4.0 J/cm^2^ CW 32 s for 3 consecutive days	Significant increase in cell proliferation was observed on day 2 and 3. Expression of collagen type 1 gene was dramatically increased on day 3.
[100]	Diode	Primary human gingival fibroblast cells (HGF cells)	2008 Saygun et al.	685 nm 2.0 J/cm^2^ CW 140 s for 1 or 2 days (one irradiation per day)	Cell proliferation was promoted in both single-dosed and double-dosed group. Single dose significantly promoted basic fibroblast growth factor (bFGF) and insulin-like growth factor-1 (IGF-1) compared to control. Double dose significantly promoted bFGF, IGF-1, and receptor of IGF-1 (IGFBP3) compared to control. None of the parameters showed significant difference between single-dosed and double-dosed group.
[101]	Diode	LPS-treated primary human gingival fibroblast cells (HGF cells)	2015 Basso et al.	780 nm 0, 0.5, 1.5, or 3.0 J/cm^2^ CW 40–240 s	Laser irradiation at 1.3 and 3.0 J/cm^2^ decreased tumor necrosis factor α (*TNFA*), *IL6*, and *IL8* gene expression, which were induced by LPS.
[102]	Diode	Primary human gingival fibroblast cells (HGF cells)	2012 Hakki et al.	940 nm 6.0, 15.0, or 20.0 J/cm^2^ CW 20 s	No significant difference was observed between laser and control group in proliferation experiment. Laser irradiation at 6, 15, and 20 J/cm^2^ significantly increased IGF, VEGF, and transforming growth factor (TGF)-β (*TGFB*) mRNA expressions. Collagen type Ⅰ mRNA expression was enhanced by 6.0 J/cm^2^ irradiation.
[103]	Diode	Primary human gingival fibroblast cells (HGF cells)	2009 Damante et al.	780 nm 3.0 or 5.0 J/cm^2^ CW 3 and 5 s	Production of bFGF was significantly higher in laser-treated group.
[104]	Diode	Primary human gingival fibroblast cells (HGF cells)	2001 Kreisler et al.	810 nm 24.64–492.8 J/cm^2^ CW 60–240 s	Laser irradiation caused significant reduction in cell numbers. Exposure time was more relevant to cell reduction than power output.
[105]	Diode	Fibroblast cell line (NIH-3T3)	2002 Pereira et al.	904 nm 3.0-5.0 J/cm^2^ CW 8–24 s for 1–6 days	Cell numbers were about 3- to 6-fold higher in laser-irradiated (3.0 and 4.0 J/cm^2^) culture compared to control. Irradiation at 5.0 J/cm^2^ had no significant effect on cell growth. 3.0 J/cm^2^ irradiation increased cell growth without affecting procollagen synthesis.
[106]	Diode	Fibroblast cell line (NIH-3T3)	2016 Sassoli et al.	635 ± 5 nm 0.3 J/cm^2^ CW	Laser irradiation inhibited TGF-β- induced fibroblast-myoblast transition. Upregulation of matrix metalloproteinase (MMP)-2 and MMP-9 and downregulation of tissue inhibitors of metalloproteinase (TIMP)-1 and TIMP-2 was shown in laser-treated group.
[108]	Diode	Human gingival fibroblast cell line (FMM1)	2004 Marques et al.	904 nm 3.0 J/cm^2^ CW 24 s	Laser irradiation caused ultrastructural changes. Procollagen synthesis was unaffected, but significant reduction in the amount of protein was observed in the medium conditioned by irradiated cells.
[76]	Nd:YAG	Fibroblast cell line (NIH-3T3)	2010 Chellini et al.	1064 nm 1.5 J/cm^2^ Pulsed 50, 70 Hz 10 s	Proliferation and cell viability was not significantly affected by laser irradiation. Type 1 collagen expression was significantly induced by 20 mJ/50 Hz laser irradiation.
[109]	Nd:YAG	Primary human skin fibroblast cells (HSF cells)	1983 Castro et al.	1060 nm 1.1 × 10^3^–2.3 × 10^3^ J/cm^2^ Pulsed	Significant reduction in DNA synthesis and collagen production was observed at 1.7 × 10^3^ J/cm^2^ irradiation. At 2.3 × 10^3^ J/cm^2^, suppression of DNA synthesis was accompanied by cell nonviability. Collagen production was inhibited, while DNA synthesis was unaffected with 1.1 × 10^3^ J/cm^2^ irradiation.
[110]	Nd:YAG	Primary human skin fibroblast cells (HSF cells)	1984 Abergel et al.	1064 nm 1.2 × 10^3^–4.7 × 10^3^ J/cm^2^ Pulsed 3–12 s	Collagen production and DNA replication was significantly decreased by laser irradiation.
[113]	Nd:YAG	Human normal epidermal keratinocyte cell line (HaCaT)/keratinocyte-conditioned medium (KCM) stimulated human dermal fibroblast cells (HDF cells)	2019 De Filippis et al.	1064 nm 2.0, 4.0, 6.0, or 8.0 J/cm^2^ Pulsed	Expression of aquaporins, filaggrin, TGase, and HSP70 was upregulated in HaCaT cells by laser irradiation. In HDF cells stimulated by KCM, reduction in MMP-1 and increase in procollagen, collagen type1, and elastin was induced by laser irradiation.
[115]	Nd:YAG	Primary human skin fibroblast cells (HSF cells)	2010 Dang et al.	532 or 1064 nm 1.5 J/cm^2^ Pulsed Irradiated twice	Both lasers upregulated collagen synthesis and gene expression of TIMPs expression, but downregulated MMPs mRNA expression at 24 and 48 h postirradiation. *TGFB* mRNA expression was promoted by 1064 nm laser. Gene expression of *HSP70* and *IL6* were promoted by 532 nm laser.
[118]	Er:YAG	Primary human gingival fibroblast cells (HGF cells)	2005 Pourzarandian et al.	2940 nm 1.68–5.0 J/cm^2^ Pulsed 20 Hz	Faster cell growth was observed in laser-treated cultures. The optimal energy was found to be 3.37 J/cm^2^.
[119]	Er:YAG	Primary human gingival fibroblast cells (HGF cells)	2005 Pourzarandian et al.	2940 nm 3.37 J/cm^2^ Pulsed 20 Hz	Laser irradiation significantly increased PGE_2_ production by HGFs. mRNA expression of *COX2* was significantly increased after laser irradiation. Inhibition of COX-2 completely suppressed PGE_2_ synthesis induced by laser irradiation.
[120]	Er:YAG	Primary human gingival fibroblast cells (HGF cells)	2015 Ogita et al.	2940 nm 1.65, 2.11, or 2.61 J/cm^2^ Pulsed 30 s	A significant cell proliferation without cell damage was shown on day 3 after irradiation. mRNA and protein expression of galectin-7 was increased after laser irradiation.
[121]	Er:YAG	Primary human gingival fibroblast cells (HGF cells)	2018 Kong et al.	2940 nm 3.6, 4.2, 4.9, 6.3, 8.1, or 9.7 J/cm^2^ Pulsed 20 or 30 Hz 20 or 30 s	Laser irradiation at 6.3 J/cm^2^ enhanced maximal cell proliferation, however, lactate dehydrogenase (LDH) release was observed on day 3 after irradiation. Laser irradiation affected cell cycle and increased proliferating cells. Transient damage was observed at 3 h. mRNA expression of *HSP70* family was increased by laser irradiation. Inhibition of thermosensory transient receptor potential channels suppressed laser-induced cell proliferation.
[122]	Er:YAG/ Er,Cr:YSGG	Human gingival fibroblast cell line (NCBI code: C-165)	2016 Talebi-Ardakani et al.	2940 nm/2780 nm 1 W/0.5 W Pulsed 10 Hz 10 or 30 s	Significant increasement in cell proliferation was shown in all laser-irradiated groups at 24 and 48 h.
[123]	Er:YAG	Primary human gingival fibroblast	2020 Tsuka et al.	2940 nm 30–150 mJ Pulsed 20 Hz	Gene expression of *COX2*, *IL1B*, *TNFA*, *BMP2*, and *BMP4* significantly increased in laser-irradiated group in a dose-dependent manner compared to the control group. *COX2* gene expression showed significant increase in the centrifugal load group compared to control group, whereas gene expression of *COX2*, *IL1B*, *TNFA*, *BMP2*, and *BMP4* was significantly higher in the laser and centrifugal loaded group than in the centrifugal load group.
[128]	CO_2_	Keloid and normal fibroblast cell line	2000 Nowak et al.	10,600 nm 2.4, 4.7, or 7.3 J/cm^2^ Pulsed 16 Hz	Population doubling time for keloid fibroblasts was shortened by 2.4 and 4.7 J/cm^2^ irradiation. Secretion of bFGF was increased by 4.7 J/cm^2^ laser irradiation in both keloid and normal group; significant in keloid group. TGF-β1 was suppressed by laser irradiation in both keloid and normal group; maximal effect occurred at 4.7 J/cm^2^.
[129]	CO_3_	Primary human dermal fibroblast cells (HDF cells)	2017 Shingyochi et al.	10,600 nm 0.1, 0.5, 1.0, 2.0, or 5.0 J/cm^2^ CW 2–40 s	Cell proliferation and migration were promoted after 1.0 J/cm^2^ irradiation. Akt, ERK, and JNK pathways were activated after 1.0 J/cm^2^ irradiation. Inhibition of Akt, ERK, or JNK pathways suppressed cell proliferation and migration induced by laser irradiation.
[130]	CO_4_	Primary fibroblast cells	1983 Apfelberg et al.	10,600 nm 5 W CW 1–3 s	CO_2_ laser does not produce a greater incidence of malignant transformation that normal controls.

**Table 3 ijms-21-09002-t003:** Summary of the effects of laser irradiation on human periodontal ligament cells.

Reference No.	Laser	Cell	Year Author	Irradiation Protocol	Major Finding
[132]	Diode	Human periodontal ligament cells (hPDLCs)	2013 Wu, et al.	660 nm 1, 2, or 4 J/cm^2^ CW 66, 132, or 264 s	Laser irradiation significantly promotes proliferation of hPDLCs. Laser irradiation enhanced the mRNA expression of osteogenic maker genes.
[133]	Diode	Periodontal ligament cell line	2014 Huang, et al.	670 nm 5, or 10 J/cm^2^ CW 2.5, or 5 s	Laser irradiation significantly decreased the protein expressions of inflammatory makers. The protein expression of osteocalcin was significantly increased in laser-irradiated cells.
[134]	Diode	Human periodontal ligament cells (hPDLCs)	2010 Mayahara, et al.	830 nm 3.82 J/cm^2^ CW 10 min	Laser irradiation significantly inhibited cyclooxygenase (*COX*)*-2* and cytosolic phospholipaseA_2_-α (*PLA2G4A*) mRNA expression.
[135]	Diode	Stretched Human periodontal ligament cells (hPDLCs)	1995 Shimizu, et al.	830 nm 346–1152 J/cm^2^ CW 0, 3, 6, or 10 min	The protein expression of prostaglandin E_2_ was significantly decreased in laser-irradiated cells. Laser irradiation was tended to decrease the protein expression of interleukin (IL) 1-β.
[136]	Diode	Human periodontal ligament cells (hPDLCs)	1997 Ozawa, et al.	830 nm 3.95–7.90 J/cm^2^ CW 10 or 20 min/day	Laser irradiation in a dose-dependent manner significantly inhibited the plasminogen activator (PA) activity in hPDLCs with stretching.
[137]	Diode	Periodontal ligament cell line	2013 Huang, et al.	920 nm 5–10 J/cm^2^ CW 2.5, or 5 s	The mRNA and protein expressions of inducible NO synthase (iNOS), TNF-a, IL-1 was decreased in lipopolysaccharide-exposed periodontal ligament cells after laser irradiation. The protein expression of pErk was significantly increased in the laser-irradiated cells compared with the nonirradiated cells.
[138]	Diode	Human periodontal ligament fibroblasts	2010 Choi, et al.	810 nm 1.97, 3.94, or 5.91 J/cm^2^ CW 10, 20, or 30 s	Proliferation, alkaline phosphatase activity, and phosphorylated ERK level were significantly increased in laser-irradiated cell at limited time point.
[139]	Diode	Human periodontal ligament fibroblasts	2020 Dehdashtizadeh, et al.	810 nm 10 J/cm^2^ CW 5 s/day	Laser irradiation reduced the protein expression of matrix metalloproteinase (MMP)-8.
[140]	Diode	Human periodontal ligament fibroblasts	2003 Kreisler, et al.	809 nm 1.96, 3.92, or 7.84 J/cm^2^ CW 75, 150, or 300 s	The proliferation rates of laser-irradiated culture were significant up to 72 h compared to control culture.
[141]	Er:YAG	Human periodontal ligament cells (hPDLCs)	2020 Lin, et al.	2940 nm 3.6, 4.2, or 6.3 J/cm^2^ Pulsed 20 Hz 75, 150, or 300 s	The proliferation, migration, and invasion abilities were induced through the upregulation of galectin-7 after laser irradiation.

**Table 4 ijms-21-09002-t004:** Summary of the effects of laser irradiation on endothelial cells.

Reference No.	Laser	Cell	Year Author	Irradiation Protocol	Major Finding
[43]	Diode	Human umbilical vein endothelial cells (HUVECs)	2015 Walter, et al.	670 nm 280 mW CW 60 s	Laser irradiation had a positive effect on cell viability.
[145]	Diode	Human endothelial cell line (HECV)	2019 Amaroli, et al.	808 nm 57 J/cm^2^ CW 1 min	Laser-irradiated cells demonstrated higher proliferation rate and increased migration ability. Laser irradiation stimulated mitochondrial oxygen consumption and ATP synthesis in HECV cell.
[146]	Diode	Human umbilical vein endothelial cells (HUVECs)	2013 Protasiewicz, et al.	808 nm 1.5, 4.5 J/cm^2^ CW 90 or 270 s	Laser irradiation diminished the pro-inflammatory and procoagulant activity of Interleukin (IL)-1β-stimulated HUVECs.
[147]	Diode	Human umbilical vein endothelial cells (HUVECs)	2003 Schindl, et al.	670 nm 2–8 J/cm^2^ CW every 48 h for a period of 6 days	Doses of between 2 and 8 J/cm^2^ induced statistically significant cell proliferation.
[148]	Diode	Human umbilical vein endothelial cells (HUVECs)	2015 Góralczyk, et al.	635 nm 2, 4, or 8 J/cm^2^ 2 times with 1-day break	Laser irradiation was significantly increased in proliferation of endothelial cells. Laser irradiation significantly reduced the concentration of soluble VEGF receptor (sVEGFR)-1 in the supernatant.
[149]	Diode	Cultured rhesus macaque choroid-retinal endothelial cells (RF/6A)	2012 Du, et al.	810 nm 45.86–76.43 J/cm^2^ CW 1 min	810 nm diode laser irradiation can induce Hsp70 hyperexpression from 12 to 18 h postirradiation in cultured choroid-retinal endothelial cells without obvious cell death.
[76]	Nd:YAG	H-end endothelial cells	2010 Chellini, et al.	1064 nm 1.5 J/cm^2^ Pulsed 50 or 70 Hz 10 s	Vinculin expression in endothelial cells could be observed in the irradiated cells. Laser irradiation did not affect cell viability.
[150]	Nd:YAG	Rat aortic endothelial cells	2017 Masuda, et al.	1064 nm 100 mJ Pulsed 5 Hz 30 s	Upregulated genes with laser irradiation were related to cell migration and cell structure (membrane stretch, actin regulation, and junctional complexes), neurotransmission, and inflammation.
[151]	Nd:YAG	Human umbilical vein endothelial cells (HUVECs)	2009 Giannelli, et al.	1064 nm 15 mJ/mm^2^ Pulsed 70 Hz 1 min	Laser irradiation attenuated intercellular adhesion molecule-1 and vascular cell adhesion molecule expression.

**Table 5 ijms-21-09002-t005:** Summary of the effects of laser irradiation on cementoblasts.

Reference No.	Laser	Cell	Year Author	Irradiation Protocol	Major Finding
[153]	Diode	Cementoblasts	2017 Bozkurt, et al.	940 nm 15, 11.4, or 105 J/cm^2^ CW 60 s/cm^2^	The mRNA expression related to only cementoblast and bone morphogenetic protein were increased in laser-irradiated cells.

**Table 6 ijms-21-09002-t006:** Summary of the effects of laser irradiation on epithelial cells.

Reference No.	Laser	Cell	Year Author	Irradiation Protocol	Major Finding
[169]	Diode	Normal human oral keratinocytes (NOKSI)	2017 Tang et al.	810 nm 1.0 and 4.0 J/cm^2^ CW 5 min	Laser treatments induced Human β defensing (HBD)-2 expression in keratinocyte cell line. transforming growth factor (TGF) β-1 pathway was activated by laser irradiation.
[170]	Diode	Human gingival epithelial cells (HGECs)	2014 Ejiri et al.	904–910 nm 5.7–56.7 J/cm^2^ Pulsed 30 kHz 1. 3, 5, or 10 min	The laser irradiation significantly increased cell proliferation and [^3^H]thymidine incorporation at various irradiation time periods. Migration of the irradiated cells was significantly accelerated compared with the nonirradiated control. Laser irradiation induced phosphorylation of MAPK/ERK at 5, 15, 60, and 120 min after irradiation. Stress-activated protein kinases/c-Jun N-terminal kinase and p38 MAPK remained unphosphorylated.
[172]	Diode	Human oral squamous epithelial carcinoma cell lines (Ca9-22 and SCC-25)	2014 Fujimura et al.	805 ± 20 nm 0.475 W Pulsed (pulse width: 100 ms) 60 s	The mRNA expression of *DEL1* was significantly upregulated by laser irradiation (*p* < 0.01). Lipopolysaccharide (LPS)-induced interleukin (IL)-6 and IL-8 expression was significantly suppressed in the LPS+laser group (*p* < 0.01). Intercellular adhesion molecule (ICAM)-1 expression was significantly higher in the LPS+laser group than in the LPS only or control groups. Compared with the control, the migration of epithelial cells was significantly increased by diode laser irradiation.

**Table 7 ijms-21-09002-t007:** Summary of the effects of laser irradiation on osteocytes.

Reference No.	Laser	Cell	Year Author	Irradiation Protocol	Major Finding
[173]	CO_2_	Osteocyte-like cells	2013 Yokose, et al.	10,600 nm 0.71, 1.42, and 2.83 J/cm^2^ CW 10 s	The mRNA expression of *Sost* was decreased and that of *Dmp1* was increased in the cells after dose-dependent laser irradiation.
[174]	Er:YAG	Osteogenic cells (osteoblast-like cells incubated for 21 days by osteoinduction)	2020 Ohsugi, et al.	2940 nm 1.5 and 3.1 J/cm^2^ Pulsed 20 Hz 30 and 60 s	The mRNA expression of *Sost* was decreased and that of *Mef2c* was increased in laser-irradiated cells.

**Table 8 ijms-21-09002-t008:** Summary of the effects of laser irradiation on osteoclasts.

Reference No.	Laser	Cell	Year Author	Irradiation Protocol	Major Finding
[176]	Diode	Osteoclasts	2006 Aihara, et al.	810 nm 9.33, 27.99, 55.98, or 93.30 J/cm^2^ CW 1, 3, 6, or 10 min/day	Laser irradiated cells showed greater amounts of staining compared to non-irradiated cells in immunohistochemistry for receptor activator of NF-kappaB (RANK). The mRNA expression of receptor activator of RANK was upregulated in low-energy irradiated cells.

**Table 9 ijms-21-09002-t009:** Summary of the effects of laser irradiation on mesenchymal stem cells.

Reference No.	Laser	Cell	Year Author	Irradiation Protocol	Major Finding
[182]	Diode	Human bone marrow-derived mesenchymal stem cells (hBM-MSCs) human adipose-derived stem cells (hASCs)	2019 Zare et al.	630 and 810 nm 0.6, 1.2, or 2.4 J/cm^2^ 10 s	Laser irradiation combined 630 and 810 nm significantly stimulated cell viability, and decreased apoptosis in hBM-MSCs and hASCs.
[183]	Diode	Mesenchymal stem cells (MSCs) isolated from femurs and tibias in rat	2012 Wu et al.	635 nm 0.5 J/cm^2^ 75 s	Microarray analysis revealed 119 differentially expressed genes after laser irradiation. mRNA expression of *Akt1*, *Ccnd1*, and *Pik3ca* were upregulated and *Ptpn6* and *Skt17b* expression were downregulated.
[184]	Diode	Mesenchymal stem cells (MSCs) isolated from femurs and tibias in rat	2008 Hou et al.	635 nm 0.5–5.0 J/cm^2^ CW 75–750 s	Laser irradiation at 0.5 J/cm^2^ stimulated MSCs proliferation. Laser irradiation at 5.0 J/cm^2^ increased VEGF and nerve growth factor secretion and dramatically facilitated the differentiation.
[33]	Diode	Human mesenchymal stromal cells (hMSCs)	2018 Tani et al.	635, or 808 nm 0.378 J/cm^2^ CW 30 s	Irradiation at 635 nm increased *Runx2* and *Alpl* mRNA expression, and expression of osteopontin and Ki67.
[185]	Diode	Mesenchymal stem cells (MSCs) isolated from femurs and tibias in male C2F1 mice	2013 Giannelli et al.	635 nm 0.3 J/cm^2^ CW 10, 26 s	Cell proliferation was increased without change of cell viability. Increase in cell proliferation was associated with the upregulation and activation of Notch-1 pathway.
[186]	Diode	Human gingival mesenchymal stem cells (HGMSCs)	2020 Feng et al.	808 nm 0.5–4.0 J/cm^2^ CW	Laser irradiation promoted cell migration but not cell proliferation. Laser irradiation at 1.0 J/cm^2^ activated mitochondrial ROS after 2 h.
[187]	Diode	Bone marrow stromal cells (MSCs) isolated from 3-old female BALB-c mice	2018 Amaroli et al.	808 nm 64 J/cm^2^ CW 60 s	Laser irradiation increased Runx2 and Osterix and decreased Pparγ protein expression. Positive areas of alkaline phosphatase and Arizarin Red S staining were significantly increased after irradiation.
[188]	Diode	Bone marrow stem cells (MSCs) isolated from young adult C57Bl/6 mice	2009 Horvat-Karajz et al.	660 nm 1.9–11.7 J/cm^2^ CW 25–75 s	Laser irradiation at 1.9 J/cm^2^ enhanced cell proliferation, although irradiation at 11.7 J/cm^2^ suppressed cell proliferation.
[189]	Diode	Stem cells from human exfoliated deciduous teeth (SHEDs)	2019 Ferreira et al.	660 nm 1–20 J/cm^2^ CW 1–28 s	Laser irradiation at 5 J/cm^2^ enhanced cell proliferation. mRNA expression of *OCT4*, *NES*, and *CD90* was increased, although that of *CD105* was decreased after irradiation at 5 J/cm^2^.
[190]	Diode	Human dental pulp stem cells (hDPSCs)	2019 Garrido et al.	660 nm 3 and 5 J/cm^2^ CW 4 and 7 s	Laser irradiation at 3 J/cm^2^ increased fibronectin expression. hDPSCs irradiated at 5 J/cm^2^ showed sign of apoptosis and necrosis.
[191]	Diode	Human dental pulp stem cells (hDPSCs)	2020 Yurtsever et al.	660 nm 0.6, or 1.6 J/cm^2^ 240 or 600 s	Laser irradiation at 1.6 J/cm^2^ increased mRNA expression of brain-derived neurotrophic factor (*BDNF*), glial cell line-derived neurotrophic factor (*GDNF*), matrix-associated protein 2 (*MAP2*), nuclear receptor-related 1 protein (*NURR1*), and dopamine transporter (*DAT*) in hDPSCs.
[192]	Nd:YAG	Human bone marrow mesenchymal stem cells (BMSCs)	2019 Wang et al.	1064 nm 2, 4, 8, or 16 J/cm^2^ 20 s	Laser irradiation at 2 and 4 J/cm^2^ promoted proliferation and osteogenesis in BMSCs. Laser irradiation at 16 J/cm^2^ suppressed proliferation and osteogenesis in BMSCs.
[193]	Nd:YAG	Horse bone marrow mesenchymal stem cells (BMSCs)	2018 Peat et al.	1064 nm 9.77 J/cm^2^ Pulsed 10 Hz 10 s	Laser-irradiated BMSCs did not show a difference in viability. Laser-irradiated BMSCs exhibited slightly lower proliferation. Interleukin (*IL*)*10* and *VEGF* mRNA expression was increased after laser irradiation.
[194]	Nd:YAG	Human adipose-derived stem cells (hADSCs)	2012 Anwer et al.	532 nm 5–45 J/cm^2^ CW 30–300 s	Laser irradiation at 5–9.2 J/cm^2^ increased cell proliferation by increasing mitochondrial activity in hADSCs. Laser irradiation at 28 and 45 J/cm^2^ decreased cell proliferation in hADSCs.
[195]	CO_2_	Human adipose-derived stem cells (hADSCs)	2017 Constantin et al.	10,600 nm 5, 9, or 10 W Pulsed 2–7 ms/shot	Laser irradiation (output power 9 W, exposure time 4 ms/shot) increased proliferation, mitochondrial ROS, the capacity to restore Δψm after rotenone-induced depolarization and the secretion of matrix metalloproteinase (MMP)-2.

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
