# Peer review of "In Vitro Cytological Responses against Laser Photobiomodulation for Periodontal Regeneration"

_ijms, 2020, doi:10.3390/ijms21239002_

Round 1
Reviewer 1 Report
The manuscript by Ohsugi et al. reviews the literature that studied the effects of laser irradiation on the cells of the periodontal tissue. A summary of the data collected from studies using a variety of laser types has been compiled in this review. Although the molecular mechanisms behind the cellular effects are still not completely understood, the authors have attempted to provide an update on the current state of understanding of the molecular effects of some of the laser types on cells of the periodontal tissue. However, the authors need to improve the manuscript for its readability and other minor corrections as below.
- While the authors provide evidence for many potential benefits of laser irradiation in this review, the feasibility of these lasers in targeting cells in periodontal disease is not well discussed. The authors need to discuss the challenges and limitations of the use of lasers to treat periodontal pathologies.
- All typos need to be corrected.
- Abstract "Suppression of inflammation by laser irradiation was observed in osteoblasts, fibroblasts, human periodontal ligament cells (hPDLCs), and endothelial cells" - The reports suggest that the laser exposure suppressed the inflammatory marker gene expression in these cell types. This should be corrected.
- Is there any effect on stem cell expansion at the injured site in the injured periodontal tissue?
Author Response
We wish to express our appreciation to the reviewer for the constructive comments and questions, which have significantly helped us to improve our manuscript. We have provided point-by-point responses to the comments and we hope that our responses address your concerns satisfactorily.
Point 1: While the authors provide evidence for many potential benefits of laser irradiation in this review, the feasibility of these lasers in targeting cells in periodontal disease is not well discussed. The authors need to discuss the challenges and limitations of the use of lasers to treat periodontal pathologies.
Response 1: Thank you for your important suggestion. According to the reviewer’s suggestion, we have added the following sentences; “Irradiation using diodes or Nd:YAG lasers is clinically feasible and can be applied in association with any periodontal procedure, since it reaches deep tissues due to its deeply penetrating wavelength. By contrast, Er:YAG and CO2 lasers, which are only superficially absorbed, are only effective on epithelial cells and connective tissue surfaces during non-surgical periodontal treatments, or exposed bone and connective tissues during periodontal surgeries. Although the purposes of periodontal treatment include anti-inflammation, tissue repair, and tissue regeneration, a single laser irradiation under a single specific irradiation condition cannot achieve all desired positive effects. Furthermore, a certain irradiation condition might have negative effects on some cells in periodontal tissues, since appropriate irradiation conditions vary with cell type.”
(page38, line738-746).
Point 2: All typos need to be corrected.
Response 2: Thank you for pointing out our typos. We have tried to corrected all typo and received English language from Editage (www.editage.com).
Point 3: Abstract "Suppression of inflammation by laser irradiation was observed in osteoblasts, fibroblasts, human periodontal ligament cells (hPDLCs), and endothelial cells" - The reports suggest that the laser exposure suppressed the inflammatory marker gene expression in these cell types. This should be corrected.
Response 3: Thank you for this insightful suggestion. According to the reviewer’s suggestion, we have modified the sentence as follows; “Laser irradiation suppressed gene expression related to inflammation in osteoblasts, fibroblasts, human periodontal ligament cells (hPDLCs), and endothelial cells.” (page1, line21-23) in the abstract.
Point 4: Is there any effect on stem cell expansion at the injured site in the injured periodontal tissue?
Response 4: It is also insightful question. MSCs are already applied periodontal regeneration. We have added following sentences; “MSCs are already clinically applied for periodontal regeneration. MSC sheets transplanted to root surfaces can induce regeneration of periodontal tissue [195]. Although further research is required to clarify the effects of laser irradiation on MSCs, laser irradiation may enhance MSCs regenerative capabilities in periodontal tissues.” in the summary of stem cells (page36, line726-730).

Reviewer 2 Report
The review paper is well written and designed. The topic is very interesting for researchers.
Authors should deepen the introduction section with the laser other uses (for e.g. doi: 10.1007/s10103-017-2384; doi: 10.1089/pho.2016.4195).
Author Response
We wish to express our appreciation to the reviewer for the constructive comments, which have significantly helped us to improve our manuscript. We have provided point-by-point response to the comments and we hope that our response addresses your concerns satisfactorily.
Point 1: While the authors provide evidence for many potential benefits of laser irradiation in this review, the feasibility of these lasers in targeting cells in periodontal disease is not well discussed. The authors need to discuss the challenges and limitations of the use of lasers to treat periodontal pathologies. The review paper is well written and designed. The topic is very interesting for researchers. Authors should deepen the introduction section with the laser other uses (for e.g. doi: 10.1007/s10103-017-2384; doi: 10.1089/pho.2016.4195).
Response 1: We thank the reviewer for this suggestion. According to the reviewer’s suggestion, we have added following sentence in the introduction; “In addition, laser irradiation is also applied to treat pressure ulcers [7] and pain associated with temporomandibular dysfunction [8].” (page1, line38-39).
We have also added some sentences in the discussion as follows; “Irradiation using diodes or Nd:YAG lasers is clinically feasible and can be applied in association with any periodontal procedure, since it reaches deep tissues due to its deeply penetrating wavelength. By contrast, Er:YAG and CO2 lasers, which are only superficially absorbed, are only effective on epithelial cells and connective tissue surfaces during non-surgical periodontal treatments, or exposed bone and connective tissues during periodontal surgeries. Although the purposes of periodontal treatment include anti-inflammation, tissue repair, and tissue regeneration, a single laser irradiation under a single specific irradiation condition cannot achieve all desired positive effects. Furthermore, a certain irradiation condition might have negative effects on some cells in periodontal tissues, since appropriate irradiation conditions vary with cell type.” (page38, line738-746).
